# LRRK2 activation controls the repair of damaged endomembranes in macrophages

Susanne Herbst[1] (ID), Philip Campbell[2] (ID), John Harvey[2], Elliott M Bernard[1],
Venizelos Papayannopoulos[3], Nicholas W Wood[2], Huw R Morris[2] & Maximiliano G Gutierrez[1,*] (ID)

## Abstract

**Cells respond to endolysosome damage by either repairing the damage or targeting damaged endolysosomes for degradation via lysophagy. However, the signals regulating the decision for repair or lysophagy are poorly characterised. Here, we show that the Parkinson's disease (PD)-related kinase LRRK2 is activated in macrophages by pathogen- or sterile-induced endomembrane damage. LRRK2 recruits the Rab GTPase Rab8A to damaged endolysosomes as well as the ESCRT-III component CHMP4B, thereby favouring ESCRT-mediated repair. Conversely, in the absence of LRRK2 and Rab8A, damaged endolysosomes are targeted to lysophagy. These observations are recapitulated in macrophages from PD patients where pathogenic LRRK2 gain-of-function mutations result in the accumulation of endolysosomes which are positive for the membrane damage marker Galectin-3. Altogether, this work indicates that LRRK2 regulates endolysosomal homeostasis by controlling the balance between membrane repair and organelle replacement, uncovering an unexpected function for LRRK2, and providing a new link between membrane damage and PD.**

**Keywords** endolysosomal damage; LRRK2; lysosomes; Parkinson's disease; tuberculosis

**Subject Categories** Immunology; Membrane & Intracellular Transport

**The EMBO Journal (2020) 39: e104494**

See also: **M Radulovic & H Stenmark** (September 2020)

## Introduction

The damage of intracellular membranes triggered by oxidative stress, internalised extracellular agents and intracellular pathogens is relevant to inflammation and neurodegeneration (Lawrence & Zoncu, 2019; Papadopoulos et al, 2019). To protect themselves from the potential detrimental consequences, mammalian cells have several mechanisms to recognise and restrict the damage of endomembranes. The ESCRT-III machinery repairs membranes that show limited damage (Radulovic et al, 2018; Skowyra et al, 2018) whereas it has been proposed that endosomes or autophagosomes repair membranes that have been extensively damaged by intracellular pathogens (Kreibich et al, 2015; Schnettger et al, 2017; Lopez-Jimenez et al, 2018). Alternatively, damaged endolysosomes are targeted to lysophagy (autophagy of lysosomes) for degradation (Settembre et al, 2011; Maejima et al, 2013). However, the mechanisms that determine whether individual damaged endolysosomes are destined for ESCRT-mediated repair or disposal are poorly understood.

Mutations and single-nucleotide polymorphisms in the kinase leucine-rich repeat kinase 2 (LRRK2) are associated with an increased risk of Parkinson's disease (PD) or Crohn's disease (Ferreira & Massano, 2017). Although the precise function of LRRK2 remains unknown, it is becoming increasingly clear that LRRK2 has a function in immunity to infection and inflammation (Herbst & Gutierrez, 2019). In different models of infection, LRRK2 has been implicated in the regulation of type I interferon (IFN) production (Hartlova et al, 2018; Weindel et al, 2020). Since organelle damage and leakage of foreign/mitochondrial DNA into the cytosol trigger type I IFN production, these observations suggest that LRRK2 could be associated with this process. However, whether endomembrane damage is implicated in LRRK2 activation and signalling is unknown.

In this study, we show that phagosome and endolysosome membrane damage via membranolytic agents or infection with *Mycobacterium tuberculosis*, *Listeria monocytogenes* or *Candida albicans* triggers LRRK2 activation. We demonstrate that LRRK2 activation leads to the phosphorylation and recruitment of Rab8A to damaged phagosomes and the subsequent recruitment of ESCRT components for membrane repair. In the absence of LRRK2 signalling, the lack of ESCRT-III recruitment on damaged endolysosomes is compensated by autophagy-mediated repair mechanisms. Importantly, we show that human primary monocyte-derived macrophages from PD patients carrying pathogenic LRRK2 mutations accumulate vesicles that are positive for the damage marker Galectin-3 with direct implications for PD pathology. These findings provide a regulatory mechanism controlled by LRRK2 that links PD pathology to endolysosome membrane damage and repair.

1 Host-Pathogen Interactions in Tuberculosis Laboratory, The Francis Crick Institute, London, UK
2 Department of Clinical and Movement Neurosciences, UCL Queen Square Institute of Neurology, UCL Movement Disorders Centre, University College London, London, UK
3 Antimicrobial Defense Laboratory, The Francis Crick Institute, London, UK
*Corresponding author. Tel: +44 (0) 203 796 1460; E-mail: Max.G@crick.ac.uk

# Results

## LRRK2 is activated by pathogen-induced phagosomal membrane damage

In order to identify the physiological stimuli that activate LRRK2 in macrophages, we infected macrophages *in vitro* with the three intracellular pathogens *M. tuberculosis*, *L. monocytogenes* and *C. albicans* and monitored LRRK2 activation by measuring the phosphorylation of the LRRK2 substrates Rab GTPases by using the Rab8A pT72 and Rab10 pT73 antibody (Lis *et al*, 2018).

We detected a significant time-dependent increase in Rab8A pT72 after infection with the pathogens confirming that infection triggers LRRK2 activation (Fig EV1A–C). On the other hand, we were not able to detect Rab10 phosphorylation after *M. tuberculosis* infection and only very low levels in response to *L. monocytogenes* and *C. albicans* infection (Fig EV1A–C). Therefore, we primarily used Rab8A pT72 phosphorylation as a read-out for LRRK2 kinase activity. Rab8A phosphorylation in infected WT macrophages was reduced in LRRK2 KO macrophages, and dependent on LRRK2 kinase activity (Fig EV1D–I). We additionally monitored LRRK2 phosphorylation on S935 which has traditionally been used as a read-out for LRRK2 kinase inhibition (Dzamko *et al*, 2010). In uninfected cells, the LRRK2 kinase inhibitors GSK2578215A and MLi-2 reduced LRRK2 pS935 phosphorylation (Fig EV1G–I). However, LRRK2 kinase inhibitors had a reduced effect on the LRRK2 pS335 levels when macrophages were infected with *M. tuberculosis* and no effect after infection with *L. monocytogenes* (Fig EV1G–I), indicating that Rab8A pT72 and LRRK2 pS935 phosphorylation are decoupled events in macrophages.

One aspect that is shared by the three pathogens is their ability to damage the phagosomal membrane and access the cytosol (Radoshevich & Cossart, 2018; Westman *et al*, 2018; Bussi & Gutierrez, 2019). We therefore tested the hypothesis that membrane damage by intracellular pathogens triggers LRRK2 activation by using mutants of *M. tuberculosis*, *L. monocytogenes,* and *C. albicans* that are unable to damage host membranes (Fig EV2). Infection of macrophages with the *M. tuberculosis* ΔRD1 mutant (Hsu *et al*, 2003), the yeast-locked *C. albicans* Δhgc1 mutant (Zheng *et al*, 2004) or the *L. monocytogenes* Listeriolysin O Δhly mutant (Radoshevich & Cossart, 2018) resulted in reduced levels of Rab8A pT72 (Fig 1A). Both LRRK2 and Rab8A localised to phagosomes containing *M. tuberculosis*, *C. albicans* and *L. monocytogenes* (Fig 1B–D), and Rab8A recruitment to phagosomes was LRRK2 kinase activity dependent (Fig EV3A–F). Additionally, an EGFP-Rab8A mutant with a threonine to alanine mutation at the reported LRRK2 phosphorylation site (EGFP-Rab8A-T72A) failed to be recruited to *M. tuberculosis, C. albicans* and *L. monocytogenes* phagosomes in macrophages (Fig EV3G–I). Strikingly, the mutants defective in their ability to damage membranes failed to recruit both LRRK2 and Rab8A (Fig 1B–D). Thus, membrane damage induced by *M. tuberculosis*, *C. albicans* or *L. monocytogenes* infection induces LRRK2 activation.

## LRRK2 is activated by sterile endolysosomal membrane damage

To validate the hypothesis that membrane damage constitutes a primary trigger for LRRK2 activation, we treated macrophages with the endolysosomal membranolytic agent L-leucyl-L-leucine methyl ester (LLOMe) (Repnik *et al*, 2017). LLOMe treatment of macrophages triggered Rab8A phosphorylation within 60 min (Fig 2A). The levels of Rab8A pT72 induced by LLOMe in WT macrophages were significantly reduced in LRRK2 KO macrophages or after LRRK2 kinase inhibition (Fig 2B and C). Both Rab8A and LRRK2 did not co-localise with the late endolysosomal marker LAMP-1 in resting conditions. Conversely, in response to LLOMe treatment, we observed an increased co-localisation of both LRRK2 and Rab8A with LAMP-1 (Fig 2D–F). We observed reduced recruitment of LRRK2 after incubation with the GSK inhibitor implying that LRRK2 kinase activity influences LRRK2 localisation (Fig 2G). Moreover, the recruitment of Rab8A to damaged endolysosomes was strikingly reduced in LRRK2 KO macrophages or after incubation with the LRRK2 kinase inhibitor (Fig 2H). We concluded that membrane damage is sufficient to trigger LRRK2 activation and Rab8A phosphorylation in macrophages.

## Rab8A is recruited to damaged endolysosomes in response to calcium signalling

We then investigated if LRRK2 was implicated in membrane repair by analysing Rab8A co-localisation with the ESCRT-III component CHMP4B, the damage marker Galectin-3, and the autophagy marker LC3B. LLOMe-induced Rab8A-positive vesicles were positive for both Galectin-3 and CHMP4B and negative for LC3B (Fig 3A and Appendix Fig S1A). Live cell imaging in macrophages expressing both EGFP-Rab8A and RFP-Galectin-3 showed that Rab8A and Galectin-3 are dynamically recruited to damaged lysosomes (Fig 3B). LLOMe treatment triggered the early formation of vesicles that initially acquired Rab8A and subsequently became positive for Galectin-3; however after 18 min, EGFP-Rab8A dissociated from the vesicles that remained RFP-Gal3 positive (Fig 3B and Movie EV1), suggesting that LRRK2-dependent Rab8A recruitment occurs at very early stages of endolysosomal damage. A time course analysis of LRRK2, Rab8A, CHMP4B, Galectin-3 and LC3B localisation after endolysosomal damage showed that LRRK2- and Rab8A-positive vesicles were visible as early as 10 min after LLOMe stimulation. The occurrence of LRRK2- and Rab8A-positive vesicles coincided with CHMP4B but not Galectin-3- or LC3B-positive vesicles which were visible after 15 min of LLOMe treatment (Fig 3C and Appendix Fig S1B). In agreement with previous reports in epithelial cells (Skowyra *et al*, 2018), we found that calcium signalling is required for the recruitment of the ESCRT machinery to damaged endolysosomes in macrophages (Fig 3D). Since Rab8A showed prominent co-localisation with the ESCRT component CHMP4B and was recruited early after damage, we tested the requirement for calcium for Rab8A phosphorylation and Rab8A and LRRK2 recruitment to damaged endolysosomes by sequestering intracellular calcium using BAPTA-AM. Rab8A pT72 phosphorylation in response to LLOMe was reduced by BAPTA-AM treatment (Fig 3E). In agreement with these findings, both LRRK2 and Rab8A recruitment to damaged endolysosomes showed a strong dependence on calcium (Fig 3F–H), indicating that LRRK2 is activated in response to intracellular calcium flux.

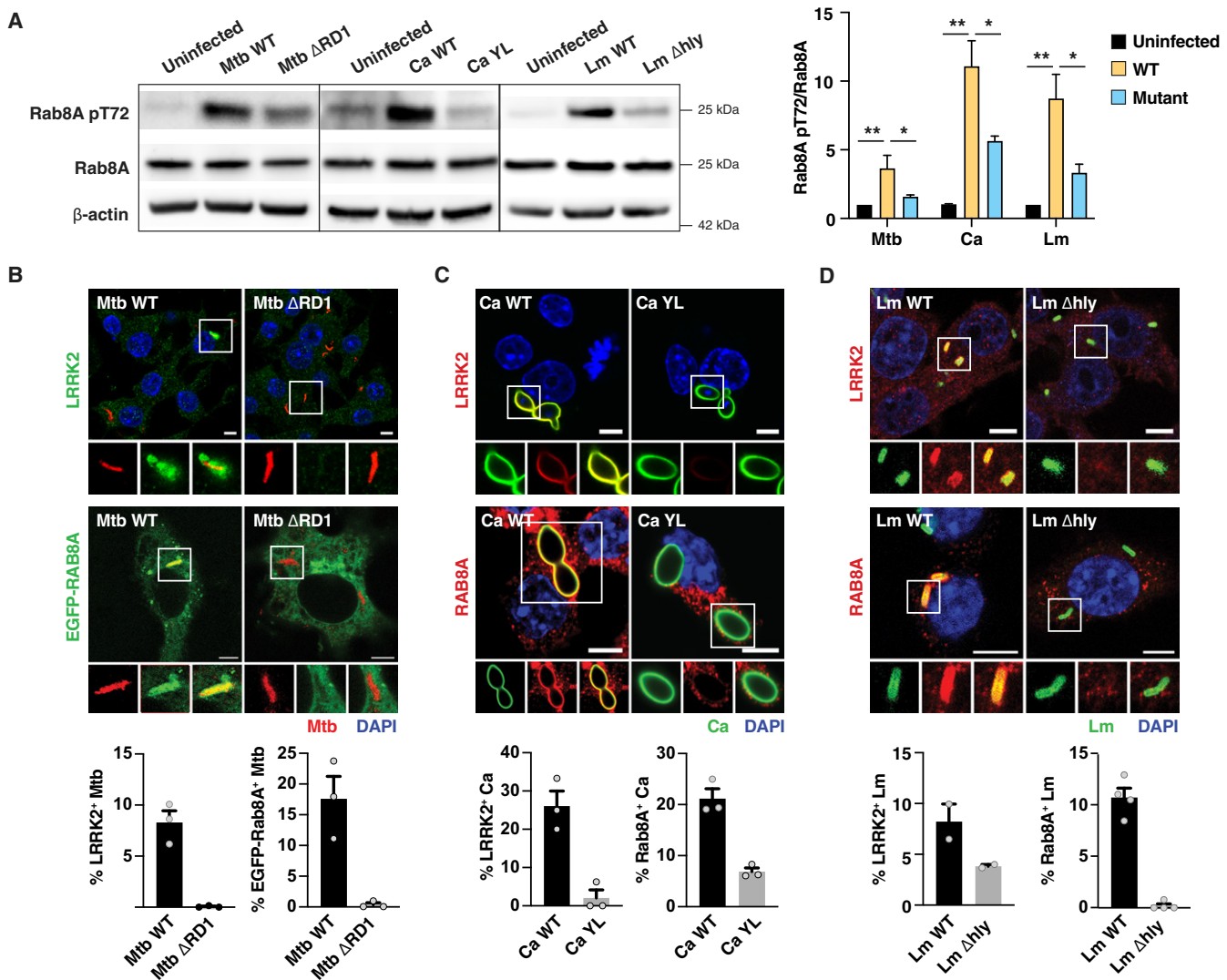

**Figure 1. LRRK2 is activated by pathogens that damage phagosomal membranes.**

A–D  RAW264.7 macrophages were infected with Mtb WT or Mtb ΔRD1 for 24 h, Ca WT or Ca YL, and Lm WT or Lm Δhly for 60 min. (A) Levels of Rab8A and Rab8A pT72 phosphorylation were analysed by Western blot and quantified by densitometry. Data represent the mean ± SEM of three independent biological experiments. One-way ANOVA followed by Dunnett's test compared to WT strains. ns = non-significant; *$P ≤ 0.05$; **$P ≤ 0.01$. (B) Recruitment of endogenous LRRK2 and EGFP-Rab8A to *Mycobacterium tuberculosis* (Mtb) was visualised by confocal microscopy. Scale bar = 5 μm. (C, D) Endogenous LRRK2 and Rab8A recruitment to (C) *Candida albicans* (Ca) and (D) *Listeria monocytogenes* (Lm) was analysed by confocal microscopy. Scale bar = 5 μm. Bottom panels show quantitative analysis of marker association to the respective pathogen. Data represent the mean ± SEM of two to four independent biological replicates.

Source data are available online for this figure.

## LRRK2 and Rab8A coordinate CHMP4B recruitment to damaged endolysosomes

Of all the LRRK2 Rab GTPase substrates we tested, Rab8A showed the most prominent recruitment to endolysosomes after damage. We also observed recruitment of Rab35 to endolysosomes after LLOMe treatment. However, Rab35-positive endolysosomes did not show the enlarged phenotype that was characteristic of Rab8A-positive endolysosomes (Fig EV4A and B). Additionally, in Rab8A KO macrophages, the Rab8A pT72 levels were significantly reduced after LLOMe treatment (Fig EV4C), indicating that Rab8A is a LRRK2 substrate after sterile damage

in macrophages. We therefore tested the role of LRRK2 and Rab8A in membrane repair of damaged endolysosomes. For this, we used high-content imaging and single cell analysis of CHMP4B- and Galectin-3-positive vesicles. The number of CHMP4B-positive vesicles after LLOMe treatment was significantly reduced in LRRK2 and Rab8A KO macrophages as well as with LRRK2 inhibitors (Fig 4A). Additionally, the overexpression of the dominant-negative mutant of Rab8A (Rab8A-T22N), which showed no recruitment to damaged endolysosomes, reduced the formation of CHMP4B-positive vesicles in cells as did the Rab8A-T72A mutant. In contrast, overexpression of the Rab8A-Q67L constitutively active mutant slightly increased the number of

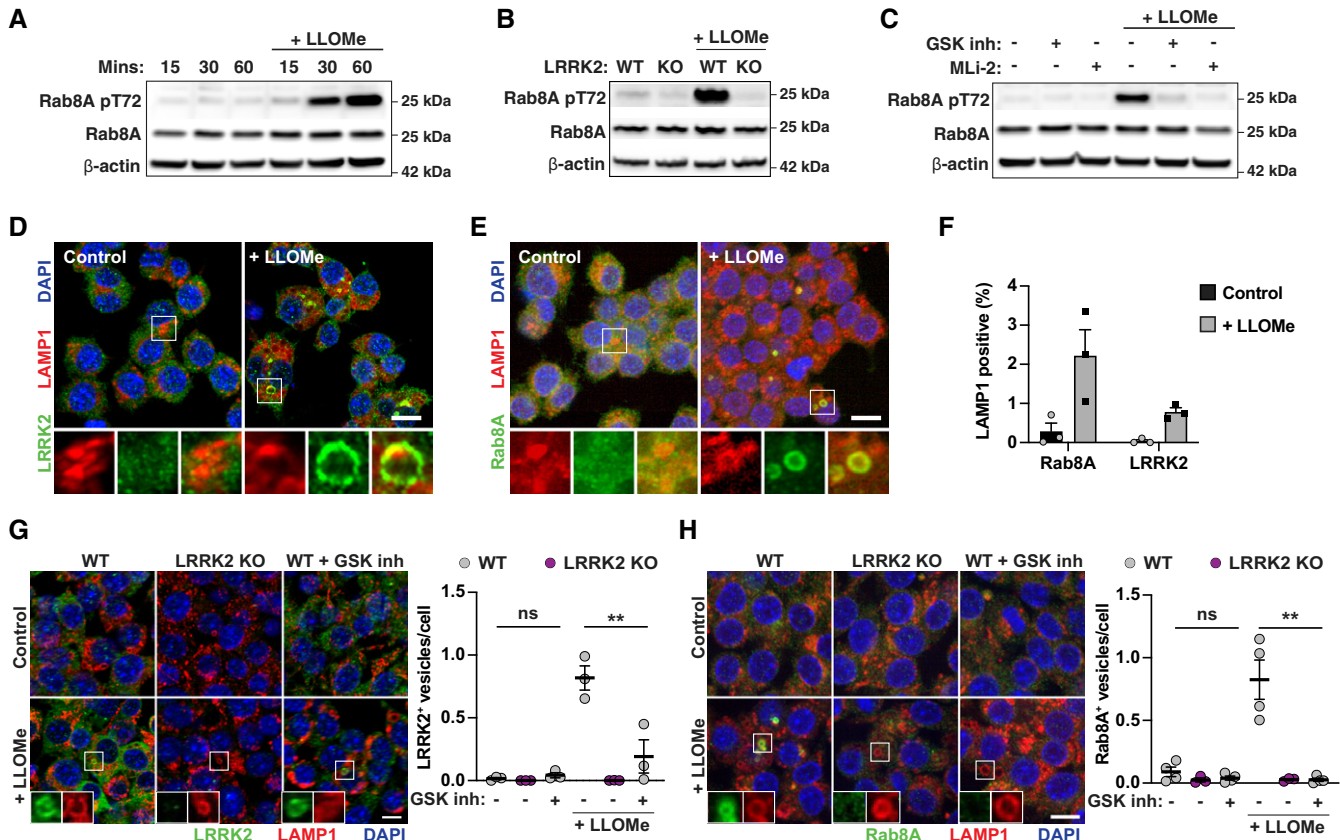

**Figure 2.   Endolysosomal damage is sufficient for LRRK2 activation.**

A       RAW264.7 macrophages were treated with 1 mM LLOMe for the indicated time, and Rab8A and Rab8A pT72 levels were analysed by Western blot.
B       WT or LRRK2 KO macrophages were treated with 1 mM LLOMe for 30 min, and Rab8A and Rab8A pT72 levels were analysed by Western blot.
C       Macrophages were pre-treated with either 1 μM GSK inh or 0.1 μM MLi-2 and then treated with 1 mM LLOMe for 30 min. Rab8A and Rab8A pT72 levels were analysed by Western blot.
D–F    RAW264.7 macrophages were treated with 1 mM LLOMe for 30 min. (D) Endogenous LRRK2 and (E) Rab8A recruitment to LAMP-1-positive compartments was visualised by immunofluorescence. Scale bar = 10 μm. (F) Quantification of (D, E). Data represent the mean ± SEM of three independent biological experiments.
G, H   LRRK2 WT or LRRK2 KO macrophages were pre-treated or not with 1 μM GSK2578215A and treated with 1 mM LLOMe for 30 min. The number of (G) LRRK2- or (I) Rab8A-positive vesicles per cell was monitored by immunofluorescence and high-content imaging. Scale bar = 10 μm. Data represent the mean ± SEM of three to four independent biological experiments. One-way ANOVA followed by Dunnett's test against the untreated WT control. ns = non-significant; **P ≤ 0.01.

Source data are available online for this figure.

CHMP4B-positive vesicles per cell (Fig EV4D and E), indicating that both LRRK2 and Rab8A are required for ESCRT-III recruitment to damaged endolysosomes.

In agreement with a role of LRRK2 kinase activity in membrane damage sensing, Galectin-3-positive vesicles were also reduced in LRRK2 and Rab8A KO macrophages or macrophages treated with

**Figure 3.   Rab8A is recruited to damaged lysosomes containing Galectin-3 and CHMP4B in response to calcium signalling.**

A       RAW264.7 macrophages were electroporated with EGFP-Rab8A and treated with 1 mM LLOMe for 30 min. Intracellular distribution of Galectin-3, CHMP4B or LC3B was visualised by immunofluorescence. Scale bar = 5 μm.
B       Macrophages were electroporated with EGFP-Rab8A and RFP-Galectin-3. Cells were then treated with 1 mM of LLOMe and monitored by live cell imaging. Snapshots at the indicated time points after LLOMe addition are shown. Scale bar = 5 μm.
C       RAW264.7 macrophages were treated with 1 mM of LLOMe, and LRRK2, Rab8A, CHMP4B, Galectin-3 and LC3B foci over time were analysed by high-content imaging. Data show the mean ± SEM of five biological replicates.
D–H    RAW264.7 macrophages were pre-treated with 10 μM BAPTA-AM for 1 h, and treated with 1 mM LLOMe for 30 min. (D) CHMP4B recruitment was monitored by immunofluorescence and high-content imaging. Scale bar = 10 μm. (E) Rab8A pT72 phosphorylation was analysed by Western blot. (F) LRRK2 and (G) Rab8A recruitment was monitored by immunofluorescence and high-content imaging. Scale bar = 10 μm. (H) Quantification of E and F. (D and H) Data represent the mean ± SEM of two to three independent biological experiments. One-way ANOVA followed by Sidak's multiple comparisons test. ns = non-significant; **P ≤ 0.01, *P ≤ 0.05.

Source data are available online for this figure.

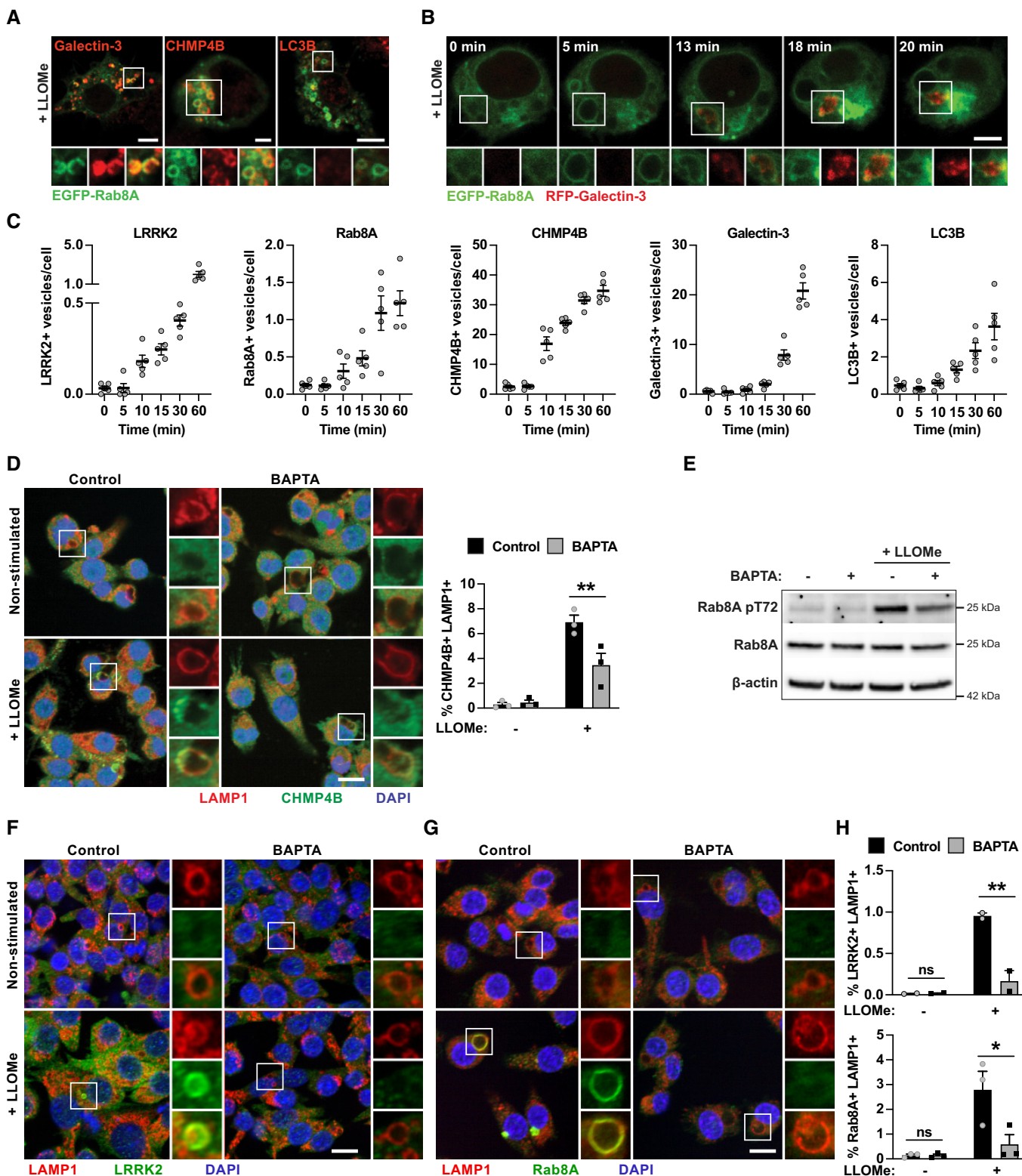

**Figure 3.**

LRRK2 inhibitor (Fig 4B). Galectin-8, another galectin that recognises membrane damage, did not co-localise with Rab8A (Fig EV5A). Moreover, LRRK2 signalling did not decrease, but in

the case of LRRK2 kinase inhibition and Rab8A KO even increased the number of Galectin-8-positive vesicles after LLOMe treatment (Fig EV5B–D). Therefore, membrane damage still occurred in the

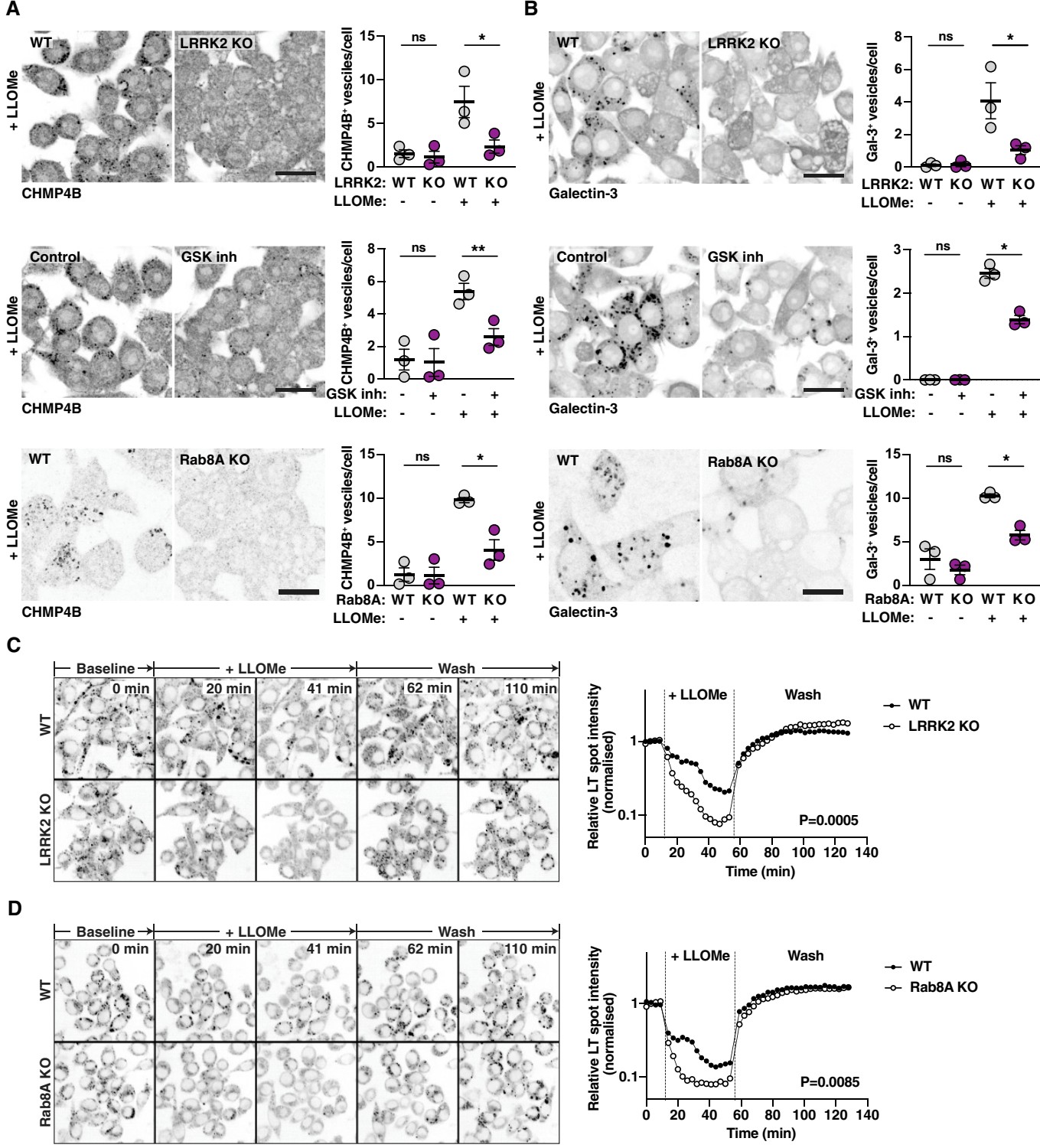

**Figure 4. LRRK2 and Rab8A coordinate the recruitment of CHMP4B to damaged endolysosomes.**

A, B  RAW264.7 WT, LRRK2 KO or Rab8A macrophages pre-treated with 1 µM GSK2578215A (GSK inh) were treated with 1 mM LLOMe for 30 min. (A) CHMP4B and (B) Galectin-3 vesicle numbers were analysed by immunofluorescence and high-content imaging. Scale bar = 20 µm. Right panels show the quantification of number of CHMP4B- or Galectin-3-positive vesicles per cell. Mean ± SEM of three independent biological experiments. ns = non-significant, *P ≤ 0.05, **P ≤ 0.01 by one-way ANOVA followed by Sidak's multiple comparisons test.

C, D  Live cell imaging of LysoTracker-positive spots in WT, LRRK2 KO or Rab8A KO macrophages treated with 1 mM LLOMe, followed by lysosomal recovery after LLOMe wash-out. One representative experiment out of three shown. Differences between slopes in the LLOMe treatment window were estimated using linear regression.

absence of LRRK2 signalling, suggesting that LRRK2 specifically affects membrane damage recognised by Galectin-3. We next used the lysosomotropic dye LysoTracker to monitor lysosomal integrity dynamics (Schnettger et al, 2017; Radulovic et al, 2018). Damaged endolysosomes in LRRK2 or Rab8A KO macrophages leaked LysoTracker faster and to a higher extent when compared to WT macrophages (Fig 4C and D). However, recovery of the lysosomal population after LLOMe removal was similar between WT and LRRK2 or Rab8A KO macrophages (Fig 4C and D), arguing that LRRK2 primarily maintains endolysosome integrity during sterile damage.

### Damaged endolysosomes are targeted to lysophagy in the absence of LRRK2 kinase activity

We next analysed the fate of damaged endolysosomes in the absence of LRRK2 signalling. The number of LC3B-positive vesicles induced by LLOMe treatment was not different between LRRK2 KO and WT macrophages (Fig 5A). In contrast, there was a significant increase in the number of LC3B-positive vesicles after LLOMe treatment in macrophages treated with the LRRK2 kinase inhibitor or in Rab8A KO macrophages (Fig 5A). We also monitored ubiquitin K63-positive vesicles as a marker for the induction of lysophagy after endolysosomal damage (Koerver et al, 2019). Consistent with an increase in lysophagy in the absence of LRRK2 signalling, LLOMe treatment resulted in an increased number of K63-positive vesicles in LRRK2 KO, LRRK2 kinase inhibitor-treated or Rab8A KO macrophages when compared to their respective controls (Fig 5B). Strikingly, in LRRK2 KO macrophages, macrophages treated with the LRRK2 inhibitor or Rab8A KO macrophages the percentage of double-positive LC3B and LAMP-1 vesicles was significantly higher (Fig 5C and D), overall indicating that damaged endolysosomes were targeted to lysophagy.

### LRRK2 regulates membrane damage-induced bacterial pathogen intracellular growth

We hypothesised that alterations in phagosome membrane repair and targeting to autophagy modulate the ability of the host cell to restrict pathogen growth. To test this, we assessed Galectin-3 and LC3B recruitment to M. tuberculosis, C. albicans and L. monocytogenes. CHMP4B was not analysed because of a cross-reaction of the antibody with bacteria. Galectin-3 recruitment to all three pathogens was significantly reduced in LRRK2 KO macrophages (Fig 6A–C), indicating that the role of LRRK2 in the endolysosomal and phagosomal damage response was similar. In addition, we observed an increase in LC3B recruitment to C. albicans and L. monocytogenes in LRRK2 KO macrophages (Fig 6B and C). As previously reported (Hartlova et al, 2018), there were no differences in autophagic targeting of M. tuberculosis in LRRK2 KO macrophages (Fig 6A). Next, we tested the role of LRRK2 in controlling pathogen replication. For this, we monitored growth by high-content imaging of M. tuberculosis, C. albicans and L. monocytogenes and their respective mutants. As previously reported, LRRK2 KO macrophages restricted M. tuberculosis growth when compared to WT macrophages (Hartlova et al, 2018). In agreement with a role for LRRK2 in membrane damage, the M. tuberculosis ΔRD1 mutant unable to induce membrane damage grew at the same level in both WT and

LRRK2 KO macrophages (Fig 6D). In contrast, both C. albicans WT and the C. albicans YL mutant grew at similar levels in WT and LRRK2 macrophages, suggesting that LRRK2 does not significantly contribute to the control of fungal pathogens (Fig 6E). On the other hand, and similarly to M. tuberculosis, LRRK2 KO restricted L. monocytogenes replication. Moreover, the L. monocytogenes Δhly mutant unable to induce phagosome damage grew at similar levels in both WT and LRRK2 KO macrophages (Fig 6F). We concluded that LRRK2 contributes to control the intracellular growth of bacterial pathogens that are able to induce membrane damage.

### Macrophages from PD patients accumulate Rab8A- and Galectin-3-positive endolysosomes

We next validated membrane damage as a trigger for LRRK2 activation in clinical samples. For this, we treated monocyte-derived macrophages from healthy controls and PD patients carrying the LRRK2 G2019S or R1441C mutation with LLOMe. Confirming our previous observations in mouse macrophages, LLOMe treatment resulted in a robust Rab8A phosphorylation which was completely inhibited by the LRRK2 kinase inhibitor MLi-2 (Fig 7A). We also observed increased Rab8A phosphorylation in macrophages from LRRK2 mutation carriers in two out of the three donors at baseline (Fig 7A). However, macrophages from LRRK2 mutation carriers did not show increased Rab8A phosphorylation in comparison with controls after LLOMe treatment implying that a maximal response had been achieved. Additionally, in both control macrophages and those carrying the LRRK2 mutations, MLi-2 completely inhibited Rab8A phosphorylation (Fig 7A). Human macrophages from healthy controls showed no Rab8A- and Galectin-3-positive vesicles in non-stimulated conditions and responded to LLOMe stimulation by Rab8A and Galectin-3 vesicle formation. However, in accordance with the elevated levels of Rab8A phosphorylation, macrophages from PD patients already showed Rab8A-positive vesicles under steady-state conditions. Unexpectedly, macrophages from PD patients also showed an increased number of Galectin-3-positive vesicles even when not stimulated with LLOMe, indicating that LRRK2 gain-of-function can drive the recruitment of Galectin-3 to endolysosomes. After LLOMe stimulation, macrophages from PD patients showed higher numbers of Rab8A-positive vesicles when compared to controls but there were no significant differences in Galectin-3-positive vesicles in this condition (Fig 7B). These results demonstrate that endolysosomal membrane damage triggers LRRK2 activation in human macrophages and show that LRRK2 gain-of-function mutations result in accumulation of vesicles that are positive for the membrane damage marker Galectin-3.

## Discussion

Here, we show that in macrophages, LRRK2 is activated after endomembrane damage, uncovering a mechanism for LRRK2 intracellular activation. Given that there are less Galectin-3-positive damaged endolysosomes in the absence of LRRK2 kinase activity, we postulate that LRRK2 contributes to membrane repair dynamics. Our data suggest that LRRK2/Rab8A signalling favours ESCRT-mediated repair over the induction of lysophagy by recruiting

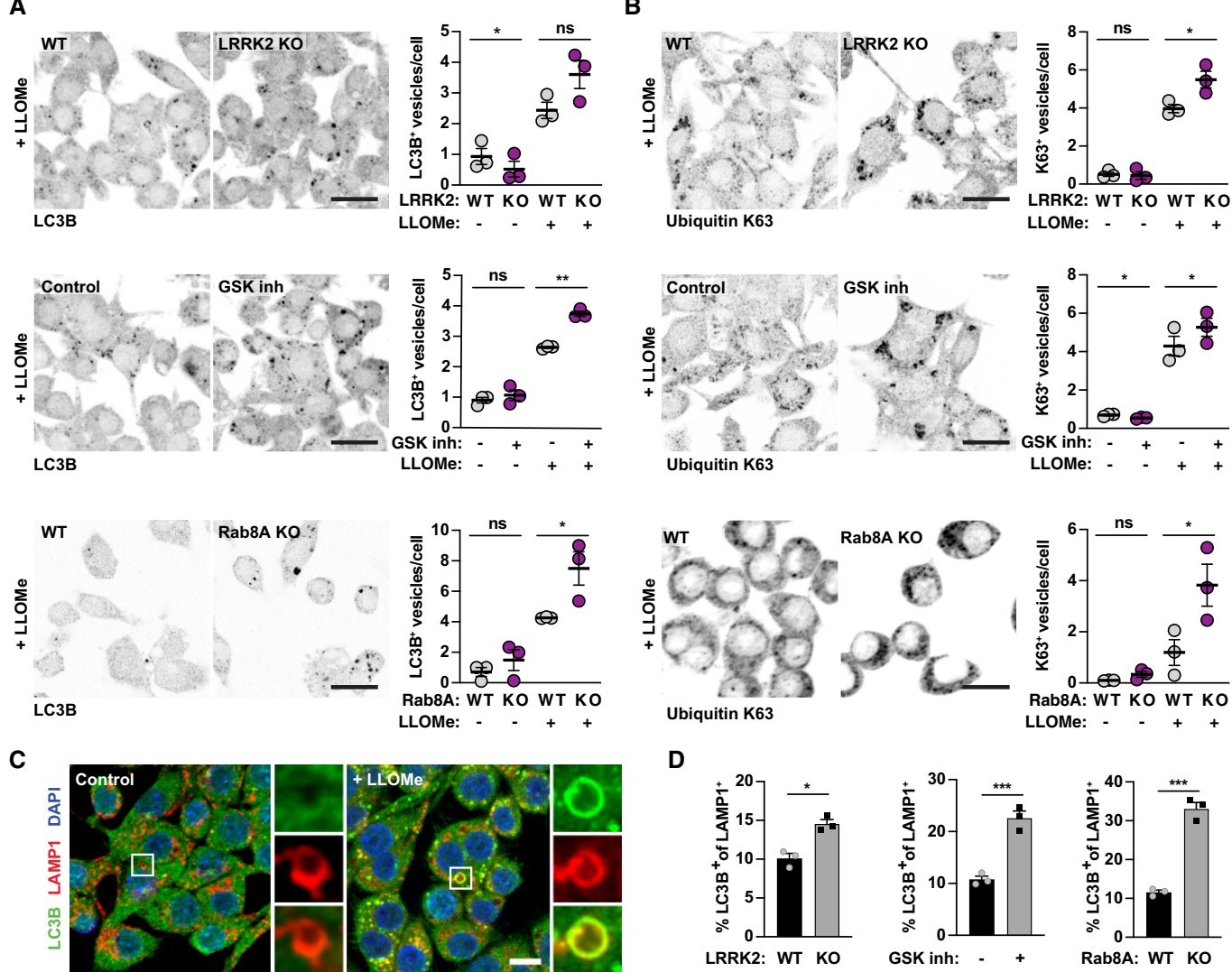

**Figure 5. Damaged endolysosomes are targeted to lysophagy in the absence of LRRK2 and Rab8A signalling.**

A, B    WT or LRRK2 KO RAW264.7 macrophages, RAW264.7 macrophages pre-treated with 1 µM GSK2578215A (GSK inh) and WT or Rab8A KO RAW264.7 macrophages were treated with 1 mM LLOMe for 30 min. (A) LC3B and (B) ubiquitin K63-positive vesicle numbers were analysed by immunofluorescence and high-content imaging. Scale bar = 20 µm. Right panels show quantification of number of positive vesicles per cell. Data show the mean ± SEM of three to four biological replicates. ns = non-significant, *P ≤ 0.05, **P ≤ 0.01 by one-way ANOVA followed by Sidak's multiple comparisons test.

C    Representative images of LC3B/LAMP-1 double-positive endolysosomes in control and LLOMe-treated macrophages. Scale bar = 10 µm.

D    Analysis of the % of LC3B/LAMP-1 double-positive endolysosomes in LLOMe-treated macrophages. Data show mean ± SEM of three independent biological experiments. *P ≤ 0.05, ***P ≤ 0.001 by Student's t-test.

ESCRT components such as CHMP4B and the damage sensor Galectin-3.

The LRRK2 signalling cascade contributes to the decision if a repair or degradation pathway for damaged endolysosome is engaged. Therefore, in the absence of LRRK2 signalling, an increase in targeting of damaged endolysosomes to autophagy likely compensates for the loss of ESCRT-III-mediated repair. Our data suggest that the previously observed induction of autophagy in the absence of LRRK2 signalling might be a consequence of the loss of alternative membrane repair mechanisms. Conversely, the overactivation of LRRK2 signalling, as occurs in a range of PD-pathogenic LRRK2

mutations, would favour attempted membrane repair over autophagy (Manzoni, 2017; Takagawa et al, 2018). It remains to be investigated if the here observed increase of Galectin-3-positive vesicles in macrophages from PD patients represent damaged vesicles or vesicles that have experienced damage but have been repaired. Therefore, it is of future interest to explore the potential consequences of ESCRT overactivation; which could be a result of LRRK2 gain-of-function mutations on endolysosomal turn-over and function. Additionally, alterations in membrane repair can modulate the ability of pathogens to grow in or escape from membrane-bound compartments. Our data demonstrate that the consequences of

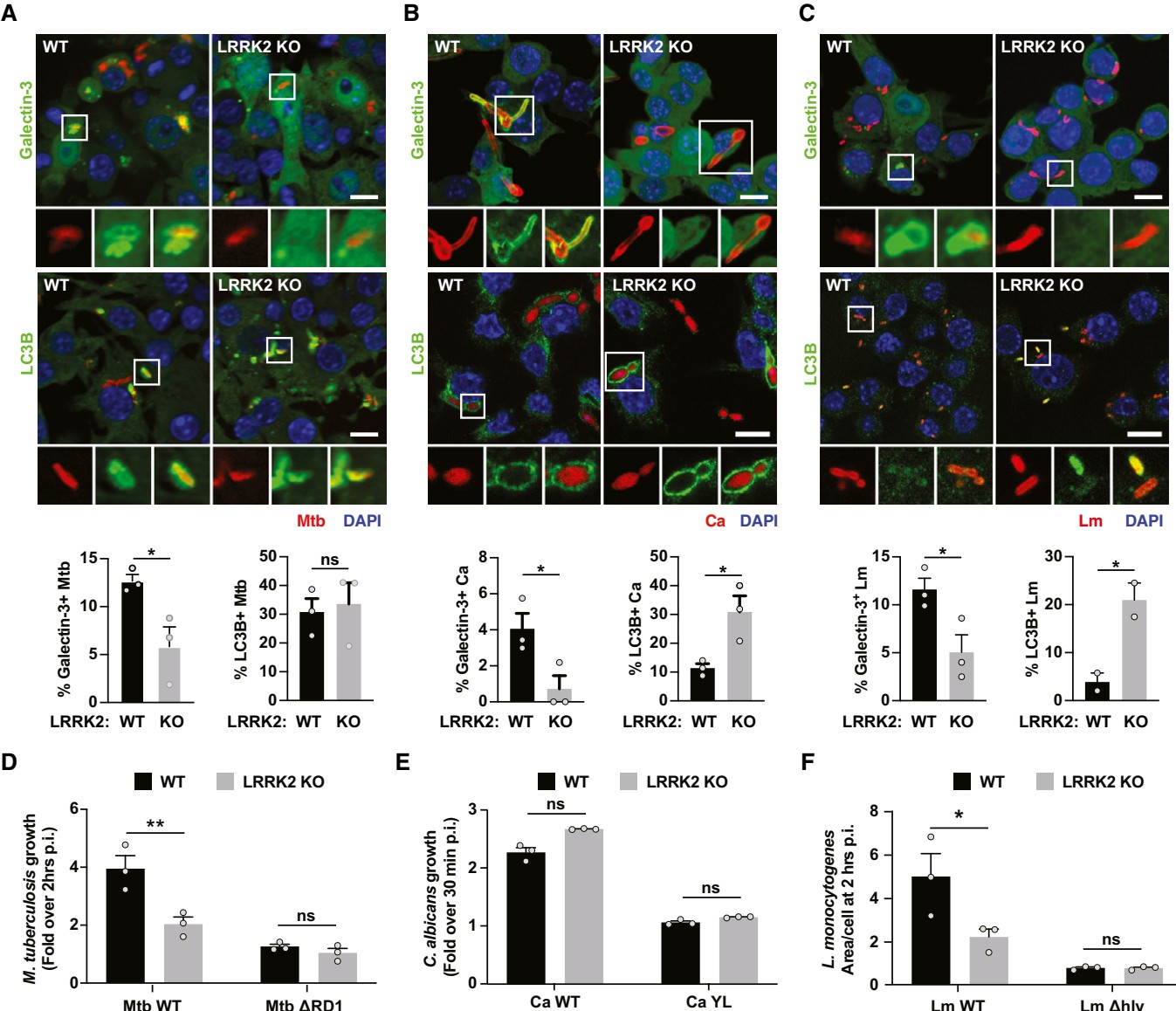

**Figure 6. LRRK2 regulates membrane damage-induced bacterial pathogen intracellular growth.**

A–C WT and LRRK2 KO RAW264.7 macrophages were infected with (A) Mtb for 24 h, (B) Ca for 60 min and (C) Lm for 60 min. Galectin-3 and LC3B recruitment was analysed by immunofluorescence and high-content imaging. Scale bar = 10 μm. Data show the mean ± SEM of two to three experiments. ns = non-significant, *P ≤ 0.05 by Student's *t*-test.

D–F WT and LRRK2 KO RAW264.7 macrophages were infected with (D) Mtb WT or Mtb ΔRD1, (E) Ca WT or Ca YL and (F) Lm WT or Lm Δhly. Growth measured for (D) 72 h and (E, F) 2 h. Data show the mean ± SEM of three independent biological experiments. ns = non-significant, *P ≤ 0.05, **P ≤ 0.01 by one-way ANOVA followed by Sidak's multiple comparisons test.

LRRK2 signalling on pathogen control vary, which is likely to be dependent on different evasion strategies. Our findings are consistent with previous reports showing that the effect of LRRK2 on pathogen immunity depends on the pathogen (Shutinoski *et al*, 2019). Our study focuses on the consequences of Rab8A phosphorylation by LRRK2 but we cannot exclude that other Rab GTPases, such as Rab10, are also phosphorylated at the same time. Unfortunately, in our system we were not able to detect Rab10 phosphorylation due to low sensitivity of the antibody (Fig EV1). We also observed recruitment of Rab35 to damaged endolysosomes; however, these endolysosomes were phenotypically different from

the enlarged Rab8A positive endolysosomes. Therefore, although we report here the downstream effects of Rab8A phosphorylation, it is likely that phosphorylation of other Rab GTPases by LRRK2 might have different functions, a research avenue that is worth exploring.

Endolysosomes will experience membrane stress throughout their life cycle, and an appropriate balance of repair and removal of those organelles is important to maintain cellular homeostasis. Additionally, as phagosomes are platforms that initiate inflammatory signalling pathways, it will be of future interest to elucidate if altered intracellular membrane integrity directly contributes to the inflammatory phenotypes that are associated with LRRK2 variants

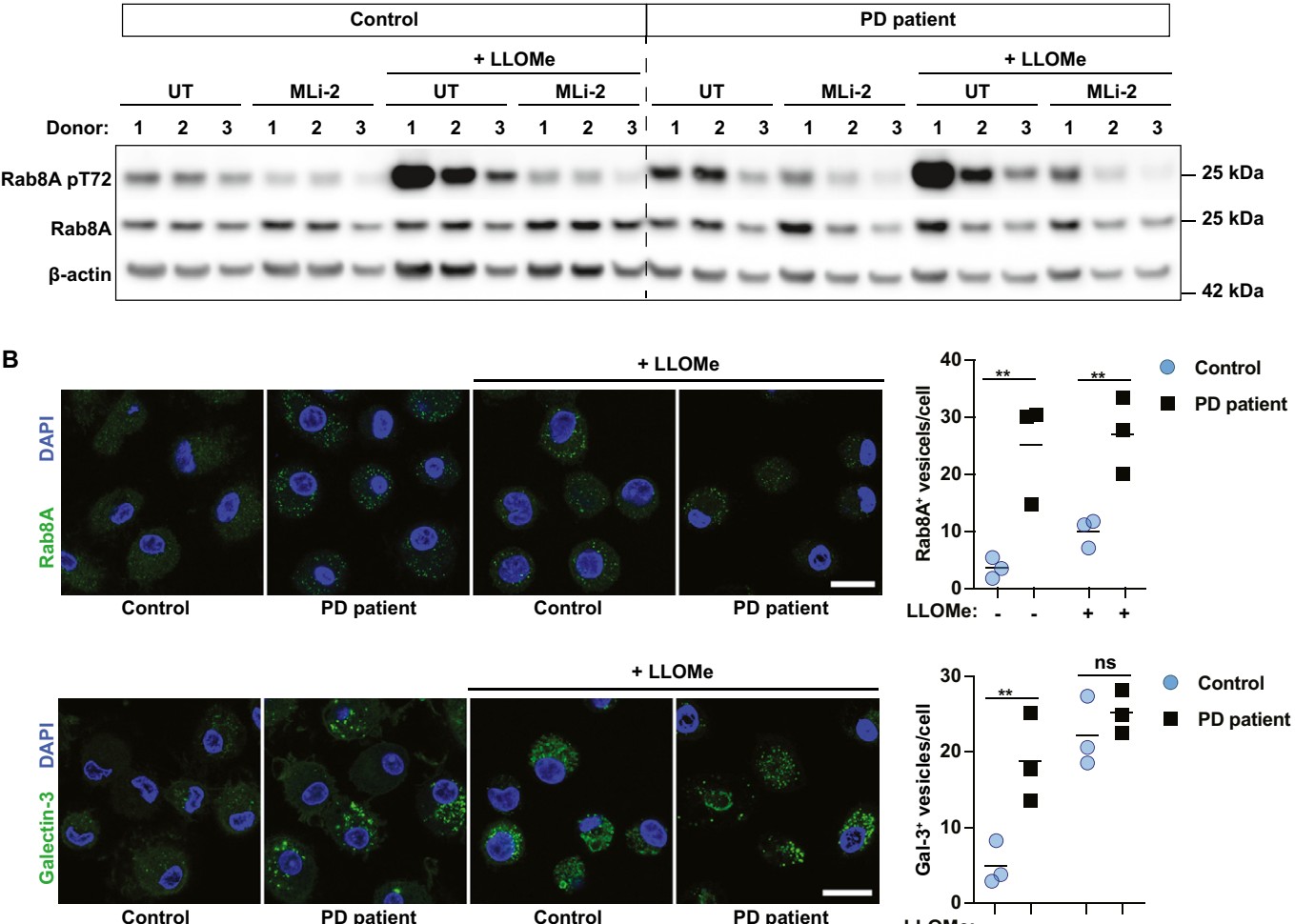

**Figure 7. Macrophages from PD patients accumulate damaged endolysosomes.**

Monocyte-derived macrophages from healthy controls and PD patients carrying the G2019S (Donor 2–3) or R1441C (Donor 1) LRRK2 mutation were pre-treated with 0.1 μM MLi-2 and treated with 1 mM LLOMe for 30 min.

A    Rab8A and Rab8A pT72 levels were analysed by Western blot.
B    Rab8A- and Galectin-3-positive vesicle numbers were analysed by immunofluorescence. Data show the mean ± SEM of biological replicates. Scale bar = 20 μm.
      ns = non-significant, **$P \leq 0.01$, by one-way ANOVA followed by Sidak's multiple comparisons test.

Source data are available online for this figure.

---

in infection, Parkinson's disease and Crohn's disease. Overall, the finding that LRRK2 is activated by membrane damage and thereby contributes to organelle maintenance has implications for immunity and inflammation in the context of both infectious and non-infectious diseases.

## Materials and Methods

### Plasmids

All DNA constructs were produced using *Escherichia coli* DH5α (Thermo Fisher Scientific) and extracted using a plasmid miniprep kit from Qiagen. EGFP-Rab8A-T72A was generated from EGFP-Rab8A

(Addgene: 86075) by site-directed mutagenesis using the Q5 SDM kit from New England Biolabs. RFP-Galectin-3 was subcloned from GFP-Galectin-3 (Addgene: 73080) into RFP-LC3B plasmid (Addgene: 21075) by restriction digest with BglII and KpnI. All plasmids were verified by sequencing. Further plasmids used in this study are pEGFP-Rab8A-T22N (Addgene: 86077), pEGFP-Rab8A-Q67L (Addgene: 86076), pEGFP-Rab3A (Addgene: 49542), pEGFP-Rab10 (Addgene: 49472), pEGFP-Rab35 (Addgene: 47424) and pTEC19 (Addgene: 30178) for transformation of *M. tuberculosis*.

### Antibodies for immunofluorescence and Western blot

Antibodies used in this study were anti-Rab8A (6975), anti-Rab10 (8127) and anti-β-actin-HRP (12262) from Cell Signaling; anti-Rab8A

pT72 (ab230260), anti-Rab10 pT73 (ab230261), anti-LRRK2 for immunofluorescence (ab133474), anti-Listeria-FITC (ab68592), anti-Candida-FITC (ab21164) and anti-LAMP1 for Western Blot (ab24170) from Abcam; anti-LRRK2 for Western Blot (N241A/34) from NeuroMab; anti-Galectin-3-AF647 (125408) from BioLegend; anti-CHMP4B (13683-1AP) from ProteinTech; anti-LC3B (PM036) from MBL; anti-Galectin-8 (AF1305) from R&D Systems; and anti-LAMP-1 for immunofluorescence (1D4B) from DSHB and anti-K63 Ubiquitin (05-1308) from Millipore.

### Ethical approval and consent

The study was approved by the Camden & Kings Cross Research Ethics Committee (REC reference: 17/LO/1166), and all samples were collected in accordance with the declaration of Helsinki (Declaration of Helsinki. Recommendations guiding doctors in clinical research. Adopted by the World Medical Association in 1964) after participants had given written informed consent.

### Human donor characterisation

All participants were of White European ethnic origin, and controls and PD patients were gender- and age-matched (Appendix Table S1). For genotyping, whole blood was taken from each participant and stored in ethylene diamine tetraacetic acid (EDTA) tubes, and DNA was extracted using the Plus XL DNA Extraction Kit (LGC Biosearch Technologies) according to the manufacturer's protocol. DNA ranging in concentration from 50 to 100 ng/μl was subsequently genotyped using the Illumina Human-Core Exome array, supplemented with custom content including 179,467 variants associated with neurodegenerative, neurological and psychiatric conditions (Blauwendraat *et al*, 2017), in particular 44 single-nucleotide polymorphisms (SNPs) associated with PD (Appendix Table S2). PD patients carried the LRRK2 G2019S or LRRK2 R1441C mutation, and none of the study participants were found to carry any other of the tested pathogenic SNPs for PD (Appendix Table S1).

### Human monocyte-derived macrophage isolation and differentiation

Monocytes were isolated from 10 ml whole blood collected in EDTA tubes. First, red blood cells were removed by centrifugation on Ficoll-Paque (28-4039-56 AD; GE Healthcare) and recovered PBMCs were washed twice in PBS-0.5 mM EDTA to remove platelets. Monocytes were isolated from PBMCs using a magnetic cell separation system with anti-CD14 mAb-coated microbeads (130-050-201; Miltenyi Biotec). CD14-positive monocytes were differentiated in complete RPMI 1640 medium (Gibco) supplemented with 10% heat-inactivated FCS and 50 ng/ml GM-CSF (130-093-862; Miltenyi Biotec). Cells were cultured at 37°C under a humidified 5% $CO_2$ atmosphere for 6 days, and the medium was replaced at day 3. On day 6, the cells were washed and detached with 0.5 mM EDTA in ice-cold PBS and plated in a 24-well plate (containing coverslips for microscopy) at a concentration of $2 \times 10^5$ cells/well in RPMI 1640 medium containing 10% heat-inactivated FCS.

### Mouse macrophages

RAW264.7 cells (ATCC Tib-71), RAW264.7 parental (ATCC SC-6003) and RAW264.7 LRRK2 KO (ATCC SC-6004) macrophages were obtained from ATCC. RAW264.7 Rab8A Crispr KO cells were generated by Synthego. The cells were grown in DMEM (Life Technologies, Gibco) supplemented with 10% heat-inactivated FCS (cDMEM) and cultured at 37°C under a humidified 5% $CO_2$ atmosphere. For immunofluorescence, $1 \times 10^5$ cells were either seeded per well on coverslips in 24-well plates or $5 \times 10^4$ cells were seeded per well in 96-well plates and for Western Blotting, $1 \times 10^6$ cells were seeded per well in 6-well plates. All cells were verified mycoplasma-free before entering tissue culture laboratories.

### *Mycobacterium tuberculosis*, *Listeria monocytogenes* and *Candida albicans* culture

The far-red *M. tuberculosis* H37Rv and H37Rv ΔRD1-E2Crimson strains were generated by transforming the bacteria with pTEC19 (Addgene #30178), a kind gift from Lalita Ramakrishnan. *Mycobacterium tuberculosis* H37Rv, and the respective ΔRD1 strains were cultured in Middlebrook 7H9 broth (M0178, Sigma-Aldrich) supplemented with 10% Middlebrook ADC (212352, BD), 0.05% Tween-80 and 0.004% glycerol. Bacteria were incubated at 37°C with constant rotation. All bacteria were grown and used at mid-exponential phase ($OD_{600}$ from 0.6 to 0.8) for infection experiments. *Listeria monocytogenes* WT and ΔhlyO were a kind gift from Pascale Cossart (Pasteur Institute, France). All *L. monocytogenes* strains were cultured in BHI at 37°C overnight. The following day, the bacteria were diluted to an OD = 0.1 and sub-cultured for a further 4 h before infection. *Candida albicans* WT and Δhgc1 were cultured in YPD at 37°C overnight. The following day, the yeasts were diluted to an OD = 0.1 and sub-cultured for a further 4 h before infection.

### Infection of macrophages with *Mycobacterium tuberculosis*, *Listeria monocytogenes* or *Candida albicans*

Bacterial and yeast cultures were pelleted by spinning for 5 min at 2,000 × *g* and washed twice with cDMEM. For *M. tuberculosis* infection, sterile 2.5- to 3.5-mm glass beads were added at a volume equal to the bacterial pellet size and the tube was vigorously shaken for 1 min to break up bacterial clumps. The bacteria were resuspended in cDMEM and centrifuged at 300 × *g* for 5 min. The supernatant was transferred into a fresh tube, and bacterial numbers were estimated by measuring the $OD_{600}$ assuming that OD 0.1 contains $1 \times 10^7$ bacteria/ml. *Mycobacterium tuberculosis* was added at a MOI = 1. The infection was allowed to proceed for 2 h, after which all cells were washed once with PBS and the medium was replaced by cDMEM. For *L. monocytogenes* infection, bacterial numbers were estimated by measuring the $OD_{600}$ assuming that OD 0.1 contains $1 \times 10^7$/ml bacteria. Macrophages were infected with a MOI = 10 for 30 min before washing once in cDMEM. For *C. albicans* infections, yeast numbers were estimated by measuring the $OD_{600}$ assuming that OD 0.1 contains $3 \times 10^6$ yeast/ml. Macrophages were infected at a MOI = 0.5 for 30 min before washing in once in cDMEM.

## Western blotting

For lysis, cells were washed once with ice-cold PBS, harvested in PBS and lysed 1× RIPA lysis buffer (Millipore) containing protease and phosphatase inhibitors (Roche and Thermo Fisher Scientific, respectively). The samples were boiled at 70°C for 10 min in LDS sample buffer and reducing agent (NuPAGE, Life Technologies) and run on a NuPAGE 4–12% Bis-Tris gel (Life Technologies). The gels were transferred onto a PVDF membrane by wet transfer. The membranes were blocked in 5% semi-skinned milk in TBS-T (TBS, 0.1% Tween-20). The membranes were incubated with primary antibodies in 5% semi-skinned milk in TBS-T at 4°C overnight and with the secondary antibodies in 5% skimmed milk in TBS-T for 1 h at room temperature. Western blots were quantified by densitometry using ImageJ.

## LRRK2 inhibitors

Macrophages were treated with GSK2578215A inhibitor (Cat# 4629, Tocris) at 1 μM or MLi-2 (Cat# 5756/10, Tocris) at 0.1 μM for 1 h before infection or treatment. The inhibitors were present during infection or LLOMe treatment and after washing LLOMe stimulated samples.

## LLOMe treatment

A 333 mM stock of LLOMe (Cat# 4000725, Bachem) was prepared in ethanol and frozen at −20°C in tightly sealed tubes. For LLOMe treatment, the medium was replaced with cDMEM containing 1 mM of LLOMe. The cells were stimulated for 30 min, after which they were either processed for downstream applications or washed twice in cDMEM for pulse-chase experiments. 0.3% ethanol in cDMEM was used in all control samples.

## Indirect immunofluorescence

Cells seeded on coverslips or in 96-well plates (Cell Carrier Ultra, Perkin Elmer) were fixed with 4% methanol-free PFA (15710, Electron Microscopy Sciences) in PBS for 24 h at 4°C for samples containing *M. tuberculosis* or for 15 min at 4°C for all other samples. The samples were quenched with 50 mM $NH_4Cl$ (A9434, Sigma-Aldrich) in PBS for 10 min at room temperature and permeabilised with 0.3% Triton X-100, 5% FCS in PBS for 20 min. For LRRK2, CHMP4B and ubiquitin K63 immunofluorescence, cells were permeabilised in ice-cold methanol for 10 min at −20°C, followed by one wash with PBS and blocking in 5% FCS in PBS for 20 min at RT.

Primary antibodies were diluted in PBS containing 5% FCS and incubated for 1 hr at RT. The samples were washed three times in PBS, and when required, the secondary antibody was added in the same way as the primary antibody (anti-rat, anti-mouse or anti-rabbit-Alexa fluor 488, Alexa fluor 568 or Alexa fluor 633, Invitrogen) for 45 min at room temperature. After three more washes with PBS, nuclear staining was performed using 300 nM DAPI (Life Technologies, D3571) in PBS for 10 min. One final wash with PBS was performed before mounting the coverslips on glass slides using DAKO mounting medium (DAKO Cytomation, S3023) or acquiring images in PBS. Images were acquired on a Leica SP8 inverted microscope or on an Opera Phenix high-content screening system (PerkinElmer).

## Plasmid electroporation and live cell imaging

RAW264.7 cells were electroporated with a total of 0.5 μg plasmid DNA using the Neon transfection system from Thermo Fisher. Cells were washed once in PBS and resuspended in buffer R. DNA was added, and cells were electroporated with one pulse of 1,680 V and 20 ms.

For analysis of the dynamic recruitment of Rab8A and Galectin-3 in LLOMe stimulated RAW264.7 cells, macrophages were plated at $1 \times 10^5$ on WillCo-dish® glass-bottom dishes. Imaging was started immediately after addition of LLOMe. Imaging was performed using a Leica TCS SP8 microscope (Leica Microsystems) equipped with AOBS, a HC PLAOP CS2 63.0 × 1.40 OIL objective and an environmental control chamber providing 37°C and 5% $CO_2$ during imaging. Images were acquired in 2-min intervals over a time frame of 2 h.

## LysoTracker leakage assay

RAW264.7 macrophages were seeded in 96-well CellCarrier Ultra plates (Perkin Elmer) at a density of $6 \times 10^4$ cells per well and rested overnight. Cells were loaded with the nuclear dye Nuclear Green LCS1 (Abcam) at a dilution of 1:5,000 and 25 nM Lyso-Tracker DND-99 (Thermo Fisher Scientific) for 10 min. The cells were washed twice in cDMEM, and the medium was replaced with cDMEM containing 25 nM LysoTracker. The cells were imaged every 3 min at 37°C, 5% $CO_2$ using an Opera Phenix high-content screening system (PerkinElmer). First, a baseline was established by imaging four time points, followed by the addition of LLOMe to a final concentration of 1 mM. After 45 min, the cells were washed twice in cDMEM, the medium was replaced with cDMEM containing 25 nM LysoTracker, and lysosomal recovery was followed for 72 min.

## Image analysis

Confocal images were analysed using the image analysis software ImageJ. Association of Rab8A, LRRK2 and Galectin-3 was quantified by creating a mask of the bacterial or yeast outline, widen the mask by 2 pixels, and measuring mean fluorescence intensity of the marker in the masked area. Markers reaching a mean fluorescence intensity over a determined threshold were counted as positive. Phagosomes of at least 100 cells per treatment condition were quantified per biological replicate.

Images acquired using the Opera Phenix high-content screening system were analysed using Harmony analysis software 4.9 (PerkinElmer). For fixed cell imaging, cells were segmented using the DAPI nuclear signal. Vesicles were identified using the "Spots" building block (global maximum), resulting in a "Number of Spots per cell" read-out parameter. For marker association to *M. tuberculosis* and *L. monocytogenes*, pathogens were identified using the "Spots" building block, whereas *Candida albicans* was identified by thresholding. In all cases, the mean fluorescence intensity of the marker of interest was measured in the masks generated and markers reaching a mean fluorescence intensity over a determined

threshold were calculated as positive. The pathogen masks were also used to quantify pathogen area per cell for growth analysis.

For LysoTracker live cell imaging, cells were segmented using the NucGreen nuclear signal. LysoTracker puncta were identified using the "Spots" building block (local maximum). The relative spot intensity of the spots for each time point was normalised to the average of the relative spot intensity before LLOMe addition, to achieve comparability between different cell lines. For each experiment, at least 2,000 cells were analysed per treatment condition.

### Statistical analysis

Statistical analysis was performed using GraphPad Prism software. The number of biological replicates and the statistical analysis performed and *post hoc* tests used can be found in the figure legends. The statistical significance of data is denoted on graphs by asterisks (*) where (*) = $P < 0.05$, (**) = $P < 0.01$, (***) = $P < 0.001$ or ns = not significant.

## Data availability

The authors declare that there are no primary datasets and codes associated with this study.

**Expanded View** for this article is available online.

### Acknowledgements
We thank Patrick Lewis, Sharon Tooze, Jeremy Carlton, Claudio Bussi and Karen Patterson for useful comments and suggestions. We thank Pascale Cossart (Institut Pasteur) for providing *Listeria* strains. We are also grateful to the Advanced Light Microscopy at the Crick for their support in various aspects of the work. This work was supported by the Francis Crick Institute (to MGG), which receives its core funding from Cancer Research UK (FC001092), the UK Medical Research Council (FC001092 to MGG and MR/R017220/1 to PC), and the Wellcome Trust (FC001092).

### Author contributions
MGG and SH conceived and designed the analysis; SH collected the data; SH performed the analysis; MGG and SH drafted the manuscript; PC, JH, NWW and HRM provided samples and contributed to analysis; and EMB and VP contributed with reagents. All authors read and provided feedback on the manuscript.

### Conflict of interest
The authors declare that they have no conflict of interest.

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
