## [Review Process File · The EMBO Journal]

LRRK2 activation controls the repair of damaged endomembranes in macrophages

Susanne Herbst, Philip Campbell, John Harvey, Elliott Bernard, Venizelos Papayannopoulos, Nicholas Wood, Huw Morris, and Maximiliano Gutierrez

DOI: [10.15252/embj.2020104494](https://doi.org/10.15252/embj.2020104494)

Review Timeline:

Submission Date:	17th Jan 20
Editorial Decision:	6th Feb 20
Revision Received:	7th Apr 20
Editorial Decision:	11th May 20
Revision Received:	31st May 20
Editorial Decision:	8th Jun 20
Revision Received:	11th Jun 20
Accepted:	15th Jun 20

Editor: Elisabetta Argenzio

Transaction Report:

Thank you for submitting your manuscript entitled "LRRK2 activation controls the repair of damaged endomembranes in macrophages" [EMBOJ-2020-104494] to The EMBO Journal. Your study has been sent to three reviewers for evaluation, whose reports are enclosed below.

As you can see, the referees consider the work potentially interesting. However, they also raise several criticisms that need to be addressed before they can support publication in The EMBO Journal. In particular, referee #1 requests you to investigate the specificity and physiological relevance of Rab8a recruitment to damaged membranes and LRRK2 activation. Also, referee #2 indicates several points that have to be addressed such as i) the trigger of LRRK2 recruitment to damaged lysosomes, ii) the role of Rab8a activity in membrane repair, iii) how phosphorylation induces Rab8a translocation to damaged endomembranes, iv) how Rab8 drives the recruitment of ESCRT proteins, and v) the role of Rab7 and CHMP4B in membrane repair and LRRK2 recruitment, respectively.

Although we are concerned that these revisions will be extremely challenging and time-consuming, we have nevertheless decided to invite a revised version of your manuscript. However, I must stress that addressing referee #1's criticisms about the physiological significance of LRRK2 activation and Rab8a recruitment to damaged membranes will be essential for the publication of your work in The EMBO Journal. In the light of that, I would be happy to discuss the requirements for the revised manuscript with you in further detail. I should also add that it is our policy to allow only a single round of revision. Therefore, acceptance of your manuscript will depend on the completeness of your responses in this revised version.

We generally grant three months as standard revision time. Competing manuscripts published during this period will not negatively impact on our assessment of the conceptual advance presented by your study.

Given the circumstances, I would understand if you were to choose not to undergo an extensive revision here and rather submit your study elsewhere. In this case, I offer you to discuss your manuscript with my colleagues at EMBO reports and Life Science Alliance.

REFEREE REPORTS

Referee #1:

In the manuscript by Herbst et al the investigators find that lysosomal injury via chemical or infectious means induces the phosphorylation of the LRRK2 kinase substrate, Rab8a. This even is indeed dependent on LRRK2 expression (via KO) and LRRK2 kinase activity (via inhibitor). There is a creative and compelling use of different bacterial strains here, as well, given that the primary cell type of interest is a macrophage cell line. Further analyses demonstrate a transient but significant recruitment of pRab8a to lysosomal membranes in this injury process, which is likewise dependent on LRRK2 and linked to the likewise recruitment of Gal3 and CHMP4B. What is provided is well conducted. Other data argues for increase lysosome injury at baseline in LRRK2 mutant macrophages and altered lysosome repair in PD but are not as well supported.

There are a few major concerns with the manuscript. The first concern is specificity. The authors rely on the pRab8a antibody which is notoriously cross-reactive with several other Rabs. The authors do explore the specificity issue, finding that Rab8a KO eliminates the signal, which was laudable. The specificity concern is whether this phosphorylation and lysosome recruitment is specific for Rab8a, and I suspect it is not. The authors falsely claim that of "all known LRRK2 Rab GTPase substrate..." when they look at four of the fourteen proposed. Furthermore, Rab10, Rab35 and Rab3a signals do appear to be recruited by LLOMe, contrary to the text description. I do

not think the response is specific and not nearly enough attention is paid here.

Moreover, the pRab10 antibody is far more sensitive and specific than the pRab8a reagent and I find it curious that it does not appear anywhere in the manuscript. I suspect the same transient response (LRRK2-dependent phosphorylation) and recruitment of Rab10 as for Rab8a as shown here, based on Fig S4A. This must be thoroughly examined both in terms of recruitment with LLOMe and the bacteria. This should have been a parallel analysis throughout. In addition, LRRK2 biochemistry is ignored, Rab8a is the sole measure of "LRRK2 activation" which should be more thoroughly examined as it appears in the title and is central to the body of work.

The second main concern is the question of biological significance. Based on the data, it would seem reasonable to conclude that lysosomal injury activates LRRK2 kinase activity which then increases the phosphorylation of Rab8a and its transient recruitment to damaged lysosomes. It also looks convincing that there is a co-recruitment of Gal3 and CHMP4B. The question that arises is, what is the physiological significance of this? Does it matter or is it epiphenomenological? If this process were critical to lysosomal repair, then we would expect far greater consequences of its inhibition or genetic knockout. What is concerning is that Gal8 levels and recruitment, another marker of lysosome injury, is not increased when this pathway is suppressed (Fig S5B, C) and unconvincing in Fig S5D. If the LRRK2/Gal3 process were critical or even influential the data would be more robust. What about the bacteria models? How does Rab8a KO affect these responses? The consequences of blocking the LRRK2/Rab8a/Gal3 system demand further investigation. It is surprising that these clever tools do not reappear after Fig 1 to establish biological significance.

I believe that the colocalization of LAMP1 and LC3B is terribly misinterpreted. The authors conclude that all instances of these proteins being in proximity is evidence of lysophagy. However, these two proteins are always in proximity in autophagolysosomes during normal autophagy - an organelle and process known to be affected by LRRK2. We know that macroautophagy will be affected by LRRK2 activity, and certainly knocking out any Rab will likely have lysosomal consequences, the interpretation that this shunts lysosomes to lysophagy is wholly unsupported in this reviewer's mind.

Lastly, the authors likely misinterpret the patient macrophage data in Fig 6. They find increased basal Gal3 and pRab8a in these cells, concluding increased lysosomal injury but not taking into account the autonomous increase in LRRK2 kinase activity in these cells that, based on their own data, ought to drive these effects. I think it premature to conclude the lysosomal injury is constitutively elevated. These data importantly support their earlier LRRK2/Rab8a/Gal3 data, however. I just think they go a bit too far in regard to the speculation around lysosome injury.

Other concerns:

- On page 3, the first paragraph has errors in referencing figures e.g. Fig SF-G and Fig. 1D-E are inaccurate.
- The EM images in Fig 4 simply show differences in protein abundance, something that could be achieved by WB. The scale and image quality does not allow for localization, and there is no co-localization which would have better leveraged the immune-EM technique. It appears the statistics used are t-tests, not appropriate for 4 groups, and I am not sure the data are normally distributed. Much of the quantification appears to me very minor differences here
- Fig 4C and D use an interesting application of LysoTracker - loss of signal after lysosome injury. However, there is a major confound in how it was done here. The authors normalize the data to 1.0 for each group, which is highly problematic. The KO of LRRK2 and Rab8a will dramatically affect the cell loading of LysoTracker and therefore the normalized rate of change. Raw data that transparently

shows the baseline effects of both KO is necessary and I suspect differences will no longer be observed when the differential baseline is taken into account. An equal magnitude loss from a one-half baseline equal twice the rate, as a percent of control.

- It is hard to see how Fig 5a LLMOe is not significant and Fig 5C is highly significant. Are these statistical conclusions biologically meaningful? What about Fig S5B vs D. It is not clear that sufficient replicates have been performed to tease out true biological differences and not just those that "reach significance"
- I often fail to see any recruitment or colocalization of Rab8a with Mtb, best example may be S3G, but there are others. The green/red signal is summed to be "yellow" but there is nothing about the Rab8a that indicates true overlap
- In general, the statistic applied not clear enough and not overly conservative. The ANOVA uses a Dunnett's for post hoc, and the rest seem to use t-test, but with adjustments for multiple comparisons. I do not understand what this means - and much of the data show substantial tails, indicating there might not be a normal distribution allowing for any parametric test

Referee #2:

Herbst et al. report that the Parkinson's disease (PD)-related kinase LRRK2 controls membrane repair of endolysosomes in macrophages. The authors found that LRRK2 is activated by pathogen-induced damage of phagosomes and by LLOMe-induced sterile damage of lysosomes. The small GTPase Rab8A was found to be phosphorylated in response to phagosome or lysosome damage, and this was accompanied by its recruitment to damaged membranes. Because recruitment was abolished upon knockout of LRRK2 or after incubation with a GSK inhibitor (which also inhibits LRRK2), the authors conclude that LRRK2-mediated phosphorylation of Rab8A causes its translocation to endomembranes in response to membrane damage. The recruited Rab8A was found to co-localize with the ESCRT-III protein CHMP4B, a mediator of lysosomal membrane repair, and knockout of LRRK2 or Rab8A inhibited recruitment of CHMP4B and the damage sensor Galectin-3 to damaged lysosomes. Knockout of Rab8A or LRRK2, or inhibition of LRRK2, led to increased localization of the autophagy protein LC3B to damaged lysosomes, indicating that damaged lysosomes are targeted for lysophagy in the absence of LRRK2/Rab8a function. Finally, the authors found that macrophages from PD patients with LRRK2 gain-of-function mutations accumulate endolysosomes that are positive for Rab8A and Galectin-3, suggesting a link between hyper-recruitment of the endolysosomal repair machinery and PD.

Even though there has been some recent progress in our understanding of the role of LRRK2 in PD, the relationship between LRRK2 mutations and PD is still enigmatic. The present findings, which are based on state-of-the-art experiments, offer a novel mechanistic explanation and are therefore of great interest. Likewise, the roles of LRRK2 and Rab8A in ESCRT-III recruitment and lysosome repair provide novel insight into a cellular mechanism of great importance. In light of this, I think the manuscript would be well suited for publication in EMBO Journal provided an adequate revision.

Major points:

1. What triggers LRRK2 recruitment to damaged lysosomes? Is it calcium efflux?
2. In general, Rab GTPases function as molecular switches that are active in their GTP-bound form. The authors should use mutagenesis to verify whether the function of Rab8A in ESCRT-III recruitment and lysosome repair depends on its GTP status.
3. It would be interesting to know how phosphorylation of Rab8A causes its translocation to damaged endomembranes. Is there an involvement of a Rab8A GEF?
4. Likewise, it remains open how Rab8A would mediate recruitment of ESCRT-III to membranes. Is any of the ESCRT proteins a Rab8A effector?
5. As Rab7 (and not Rab8) is found on the late endosomes/lysosomes under steady-state conditions, the authors should check if Rab7 is recruited to damaged lysosomes and if it is involved in the repair process.
6. The manuscript would highly benefit if data were presented in the format of scatterplots instead of bar graphs providing information about individual data points (average) from each experiment. In line with this, in Fig 2 G,I the authors should show the mean of 3 individual experiments, not the data points from one representative experiment.
7. In Fig 4, Is LRRK2 recruited to damaged lysosomes upon CHMP4B KD?
8. It has been shown that CHMP4B is recruited to lysosomes only minutes after injury (Radulovic et al, Skowyra et al., 2018). Does LRRK2 follows the same dynamics? The earliest point presented in the manuscript is 20 min after damage.
9. It is also evident from the literature that TSG101 and ALIX are necessary for the repair of damaged lysosomes. It would be of great interest to determine if LRRK2 KO cells show impaired recruitment of TSG101 and ALIX to damaged lysosomes. These questions should be preferably addressed using live-cell imaging.

Minor points:

1. Fig 1 A: In the quantification graph, instead of 'KO' maybe 'mutant' would be better'.
2. Fig 2 D,E/ Fig 3 A,B: show control images without LLOMe treatment.
3. Fig 2 G,I: include insets and zoom in for better visualization.
4. Include LLOMe catalog number.
5. Fig 1 B,C,D: Put the square around the area that is being zoomed in on the microscopy images.
6. Figures are not correctly annotated in the last paragraph in the Results part under 'LRRK2 is activated by phagosomal and endolysosomal membrane damage' section.

7. In Figure legends for Fig 3 C,D: indicate for how long cells were pre-treated with 10 μ M BAPTA-AM.

8. In Fig S 4B it is difficult to understand what the authors mean with "levels of Rab8 pT72 in Rab8A KO macrophages are significantly reduced after LLOMe treatment". How can Rab8A be phosphorylated in Rab8A KO cells?

9. The authors should check the citation format.

Referee #3:

The authors report for the first time that the LRRK2 kinase controls the repair of intracellular pathogen- (Mtb, Lm and Ca) or chemical-induced damaged endomembranes (phagosomes and endolysosomes) through activating Rab8a and recruiting the ESCRT complex, thus preventing damaged endomembrane targeting to lysophagy.

This is an important discovery in the field of cell biology and infection biology, that is supported by a nice combination of approaches, including cell imaging, and various pathogens. Importantly, the study also includes primary material (monocyte-derived macrophages) from patients carrying LRRK2 mutations.

Major comments

The conclusions are well supported by the data. It would have been nice to study the behavior of endomembranes in infected macrophages derived from PD patients' monocytes. Has this been done, please at least discuss.

Minor comments

It is shown that Rab8-T72A is not recruited to the M. tuberculosis phagosome. Does this also hold true for Lm and Ca infected cells?

In addition to Rab8, Rab35 seems to show an increased recruitment to endolysosomes after damage; is that true, if so please discuss.

References to the figures in the first Results paragraph are inverted or mixed (Fig 2A instead of 1A, 2D-F instead of 1D-F, 2G-H instead of S2G-H etc.) Please fix.

EMBOJ-2020-104494

Point by point reply to editor and reviewers

Referee #1:

The first concern is in specificity. The authors rely on the pRab8a antibody which is notoriously cross-reactive with several other Rabs. The authors do explore the specificity issue, finding that Rab8a KO eliminates the signal, which was laudable. The specificity concern is whether this phosphorylation and lysosome recruitment is specific for Rab8a, and I suspect it is not.

We agree that the Rab8A antibody cross reacts with other Rab GTPases. We focused this work on pRab8A because it is the only one giving us a clear phenotype with regards to membrane damage. We now show the Western Blot for pRab10 after infection with the 3 pathogens (new Fig S1) where it is clear that phosphorylation of Rab10 does not follow the same pattern. Interestingly, it seems that Lm also induced pRab10. We are not claiming here the LRRK2/Rab8A is unique and it is very likely other Rab GTPases are phosphorylated and have other functions. In the case of infection, it is likely that receptor activation plays an important role. We now discussed this aspect.

The authors falsely claim that of "all known LRRK2 Rab GTPase substrate..." when they look at four of the fourteen proposed. Furthermore, Rab10, Rab35 and Rab3a signals do appear to be recruited by LLOMe, contrary to the text description. I do not think the response is specific and not nearly enough attention is paid here.

We agree that we did not look at all Rab GTPases for technical reasons, only a subset that is known to be phosphorylated by LRRK2. Rab3A and Rab10 are actually not recruited, we believe that the provided images are clear, as reviewer 3 commented. We have now changed this in the text to reflect better the Rab GTPases that were actually tested in this study.

Moreover, the pRab10 antibody is far more sensitive and specific than the pRab8a reagent and I find it curious that it does not appear anywhere in the manuscript. I suspect the same transient response (LRRK2-dependent phosphorylation) and recruitment of Rab10 as for Rab8a as shown here, based on Fig S4A. This must be thoroughly examined both in terms of recruitment with LLOMe and the bacteria. This should have been a parallel analysis throughout.

The reviewer is right that the pRab10 antibody is more specific for western blot. Unfortunately, the Rab10 antibody is not good for immunofluorescence (only recently there is one available) and the pRab10 is not as sensitive as the pRab8 antibody for WB in mouse cells (at least in our hands). However, we show there is no recruitment of Rab10 after LLOMe treatment by IF. To further strengthen this data, we now also include membrane fractionation of LLOMe stimulated cells blotted for endogenous Rab10 (**new Figure S4B**) which shows increased membrane fractions of Rab8A but not Rab10. Given these results and the technical limitations of the antibodies, we did not attempt any further analysis of Rab10 for the pathogens. We don't claim that Rab8A is the only LRRK2

substrate phosphorylated in response to membrane damage however decided to focus on the Rab8A/LRRK2 relationship.

In addition, LRRK2 biochemistry is ignored, Rab8a is the sole measure of "LRRK2 activation" which should be more thoroughly examined as it appears in the title and is central to the body of work.

We understand the concern. For many years, the readout for LRRK2 kinase inhibition was pS935. We have now included this analysis in **new Figure S1**. Clearly, pS935 is inhibited by the inhibitors but it does not entirely reflect LRRK2 activation. Our data show that these two biochemical events are dissociated.

The second main concern is the question of biological significance. Based on the data, it would seem reasonable to conclude that lysosomal injury activates LRRK2 kinase activity which then increases the phosphorylation of Rab8a and its transient recruitment to damaged lysosomes. It also looks convincing that there is a co-recruitment of Gal3 and CHMP4B. The question that arises is, what is the physiological significance of this? Does it matter or is it epiphenomenological? If this process were critical to lysosomal repair, then we would expect far greater consequences of its inhibition or genetic knockout.

We show here that autophagy compensates for the lack of this repair mediated by Rab8A. So, only by blocking autophagy in cells lacking Rab8 or LRRK2 we can test whether the phenotype is strong (e.g. cell death). This was the ongoing work that had to be stopped by the COVID-19 situation explained in the letter to the editor. However, we believe we address the biological significance by looking at pathogen control (see below).

What is concerning is that Gal8 levels and recruitment, another marker of lysosome injury, is not increased when this pathway is suppressed (Fig S5B, C) and unconvincing in Fig S5D. If the LRRK2/Gal3 process were critical or even influential the data would be more robust. What about the bacteria models? How does Rab8a KO affect these responses? The consequences of blocking the LRRK2/Rab8a/Gal3 system demand further investigation. It is surprising that these clever tools do not reappear after Fig 1 to establish biological significance.

Are Gal8 and Gal3 labelling the same subset of damaged endolysosomes? Not all damaged endolysosomes are recognised by both Gal8 and Gal3. There is a temporal control (Jia et al., 2018; Jia et al., 2020) but this is unlikely to explain our results. We believe that is possible that Gal3 and Gal8 recognise different types of damage in different vesicles, and our work shows some evidence supporting this idea. Clearly more work is required to explore further this hypothesis. We do not completely understand what this reviewer refers to with "biological significance". There are very important biological consequences of membrane damage and repair, including cell death (although not always) and we showed that it is important for the control of Mtb. We agree that it is important to show here the link between immune control and membrane damage and now added the infection with Mtb, Listeria and Ca in WT and LRRK2 KO macrophages (**new Figure 6**).

I believe that the colocalization of LAMP1 and LC3B is terribly misinterpreted. The authors conclude that all instances of these proteins being in proximity is evidence of lysophagy. However, these two proteins are always in proximity in autophagolysosomes

during normal autophagy - an organelle and process known to be affected by LRRK2. We know that macroautophagy will be affected by LRRK2 activity, and certainly knocking out any Rab will likely have lysosomal consequences, the interpretation that this shunts lysosomes to lysophagy is wholly unsupported in this reviewer's mind.

We understand the concern and apologise for the confusion. In the original Figure 5D-E, what we measured was the percentage of the LC3+ vesicles induced by LLOMe that colocalise with LAMP-1. In untreated cells the number of LC3+ structures are very low (**old Figure 5A-C**) and increased after damage induction with LLOMe. The analysis does not count proximity, it rather measures vesicles that contains both markers as other groups have used in the past (Maejima et al., 2013). We and others interpreted that after membrane damage, endolysosomes LAMP1+ are targeted to autophagy. To provide further evidence, we now analysed Ubiquitin K63 another marker of lysophagy in Rab8A and LRRK2 KO background and macrophages treated with LRRK2 kinase inhibitor (Koerver et al., 2019). Data show that after damage, the number of K63 + vesicles is significantly higher in Rab8A KO, LRRK2 KO and LRRK2 inhibitor treated macrophages (**new Figure 5**).

Lastly, the authors likely misinterpret the patient macrophage data in Fig 6. They find increased basal Gal3 and pRab8a in these cells, concluding increased lysosomal injury but not taking into account the autonomous increase in LRRK2 kinase activity in these cells that, based on their own data, ought to drive these effects. I think it premature to conclude the lysosomal injury is constitutively elevated. These data importantly support their earlier LRRK2/Rab8a/Gal3 data, however. I just think they go a bit too far in regard to the speculation around lysosome injury.

We understand the concern and now changed the discussion regarding the interpretation of these data. We did not conclude increased lysosomal injury, but an increase in LRRK2 kinase-driven effects. As such we would propose increased ESCRT-mediated repair and Galectin-3 recruitment which potentially impedes autophagy-driven removal of non-functional lysosomes. However, we were not able to test this hypothesis due the limited nature of patient samples. We will rewrite this section to clarify and shift the conclusion to the discussion section. We understand the concern and now changed the discussion regarding the interpretation of these data and moved the conclusion to the discussion.

On page 3, the first paragraph has errors in referencing figures e.g. Fig SF-G and Fig. 1D-E are inaccurate.

Thanks, we have now changed this.

The EM images in Fig 4 simply show differences in protein abundance, something that could be achieved by WB. The scale and image quality does not allow for localization, and there is no co-localization which would have better leveraged the immune-EM technique.

The images shown in Figure 4 are indirect immunofluorescence images shown in black and white for better visualisation. It shows the number of vesicles positive for a specific marker under the different conditions.

It appears the statistics used are t-tests, not appropriate for 4 groups, and I am not sure the data are normally distributed. Much of the quantification appears to me very minor differences here

The data is normally distributed. We have now reanalysed the data using ANOVA and post-hoc test.

Fig 4C and D use an interesting application of LysoTracker - loss of signal after lysosome injury. However, there is a major confound in how it was done here. The authors normalize the data to 1.0 for each group, which is highly problematic. The KO of LRRK2 and Rab8a will dramatically affect the cell loading of LysoTracker and therefore the normalized rate of change. Raw data that transparently shows the baseline effects of both KOs is necessary and I suspect differences will no longer be observed when the differential baseline is taken into account. An equal magnitude loss from a one-half baseline equal twice the rate, as a percent of control.

We understand this concern. Please see below the raw data.

Figures for Referees not shown.

It is hard to see how Fig 5a LLM0e is not significant and Fig 5C is highly significant. Are these statistical conclusions biologically meaningful? What about Fig S5B vs D. It is not clear that sufficient replicates have been performed to tease out true biological differences and not just those that "reach significance"

The analysis was done using high content imaging, which is a very powerful approach to analyse biological differences in this context (Jia et al., 2018; Jia et al., 2020) We have performed these experiments at least 3 times (biological replicates and counted at least 1,000 cells per condition. We now add this additional information in the text and legends.

I often fail to see any recruitment or colocalization of Rab8a with Mtb, best example may be S3G, but there are others. The green/red signal is summed to be "yellow" but there is nothing about the Rab8a that indicates true overlap

We agree that despite a clear colocalization in Figure 1, Figure S3-G was less clear. We have now replaced that image for a most representative one.

In general, the statistic applied not clear enough and not overly conservative. The ANOVA uses a Dunnett's for post hoc, and the rest seem to use t-test, but with adjustments for multiple comparisons. I do not understand what this means - and much of the data show

substantial tails, indicating there might not be a normal distribution allowing for any parametric test

The data is normally distributed and following the reviewer's comments we now used ANOVA and post hoc test. We have now clarified this in the material and methods section.

Referee #2:

Even though there has been some recent progress in our understanding of the role of LRRK2 in PD, the relationship between LRRK2 mutations and PD is still enigmatic. The present findings, which are based on state-of-the-art experiments, offer a novel mechanistic explanation and are therefore of great interest. Likewise, the roles of LRRK2 and Rab8A in ESCRT-III recruitment and lysosome repair provide novel insight into a cellular mechanism of great importance. In light of this, I think the manuscript would be well suited for publication in EMBO Journal provided an adequate revision.

Many thanks for the positive and constructive comments.

What triggers LRRK2 recruitment to damaged lysosomes? Is it calcium efflux?

We do think it is calcium efflux and we have shown that in the original manuscript and now in the **new Figure 3**. In these experiments, we show that BAPTA significantly reduces the association of both LRRK2 and Rab8A to damaged endolysosomes.

In general, Rab GTPases function as molecular switches that are active in their GTP-bound form. The authors should use mutagenesis to verify whether the function of Rab8A in ESCRT-III recruitment and lysosome repair depends on its GTP status.

This is a very good point. We now include data in WT macrophages expressing Rab8A-Q67L, Rab8A-T22N and Rab8A-T72A (**new Figure S5**). Our data show that the expression of the constitutively active mutant Rab8A-Q67L slightly increased the number of CHMP-4 vesicles after damage whereas the expression of the dominant negative form Rab8A-T22N significantly reduced CHMP-4 vesicle formation after damage (**new Figure S5**). We have now discussed this in the text.

It would be interesting to know how phosphorylation of Rab8A causes its translocation to damaged endomembranes. Is there an involvement of a Rab8A GEF?

This is a very good point too, we believe this is a time-consuming experimental work that is beyond the scope of this paper. It is definitely the natural next step of this project.

Likewise, it remains open how Rab8A would mediate recruitment of ESCRT-III to membranes. Is any of the ESCRT proteins a Rab8A effector?

This is another very good point, we believe this is a time-consuming experimental work that is beyond the scope of this paper. The literature on the effect of Rab GTPase phosphorylation is actually confusing and we believe that looking at effectors binding (or not) with pRab8A in macrophages is critical. However, we feel this beyond the scope of this work.

As Rab7 (and not Rab8) is found on the late endosomes/lysosomes under steady-state conditions, the authors should check if Rab7 is recruited to damaged lysosomes and if it is involved in the repair process.

Very interesting point. Rab7 is also required for autophagy (Gutierrez et al., 2004) and that might complicate the interpretation of these experiments. We did not focus on Rab7 as it is not a direct LRRK2 substrate and is phosphorylated by other kinases (such as TBK1 and LRRK1) and therefore beyond the scope of this study.

The manuscript would highly benefit if data were presented in the format of scatterplots instead of bar graphs providing information about individual data points (average) from each experiment. In line with this, in Fig 2 G,I the authors should show the mean of 3 individual experiments, not the data points from one representative experiment.

We agreed with this has now changed the presentation of the data accordingly (new Figures 1, 2, 3 and 6).

In Fig 4, Is LRRK2 recruited to damaged lysosomes upon CHMP4B KD?

We have not done this particular experiment but tried to. Knockdown of CHMP4B in macrophages is very difficult to achieve mostly because reducing the levels of CHMP4B induces toxicity. However, we believe that the LRRK2 KO data establishes directionality of the pathway so we believe this is perhaps not critical for the manuscript.

It has been shown that CHMP4B is recruited to lysosomes only minutes after injury (Radulovic et al, Skowyra et al., 2018). Does LRRK2 follows the same dynamics? The earliest point presented in the manuscript is 20 min after damage.

This is a very important question. Unfortunately, doing live cell studies with LRRK2 are very challenging for various reasons. Mostly because this kinase is very big and overexpression is difficult to achieve as well as reports showing mislocalisation of LRRK2 tagged with fluorescent proteins. We had tried unsuccessfully to produce endogenously tagged LRRK2 but now trying other alternatives. However, fixed cell imaging showed that Rab8A localisation is entirely dependent on LRRK2, and Rab8A itself was recruited within 5 minutes after the addition of LLOMe when imaged live (Figure 3B). Therefore, we think that our results are in line with the current knowledge on ESCRT recruitment to damaged lysosomes.

It is also evident from the literature that TSG101 and ALIX are necessary for the repair of damaged lysosomes. It would be of great interest to determine if LRRK2 KO cells show impaired recruitment of TSG101 and ALIX to damaged lysosomes. These questions should be preferably addressed using live-cell imaging.

These are experiments we agreed were important and started to do but because of the COVID-19 situation we are not able to perform in the next 3 months (at best).

Fig 1 A: In the quantification graph, instead of 'KO'; maybe 'mutant'; would be better;

We have changed it now. Thanks.

Fig 2 D,E/ Fig 3 A,B: show control images without LLOMe treatment.

We are now showing the control images (**new Figure 2 and new Figure 3**) Figure 3AB (new Figure S4). For Figure 3B, t=0 shows cells just after LLOMe addition and therefore serves a control.

Fig 2 G,I: include insets and zoom in for better visualization.

We now included the zooms in.

Include LLOMe catalog number.

This information was included in the table provided and now included in text as well.

Fig 1 B,C,D: Put the square around the area that is being zoomed in on the microscopy images.

Thank you for spotting this. We have now included the zoomed in areas.

Figures are not correctly annotated in the last paragraph in the Results part under 'LRRK2 is activated by phagosomal and endolysosomal membrane damage' section.

We have now correctly annotated the figures.

In Figure legends for Fig 3 C,D: indicate for how long cells were pre-treated with 10 μ M BAPTA-AM.

Cells were treated for 1 hour; this information is now included.

In Fig S 4B it is difficult to understand what the authors mean with "levels of Rab8 pT72 in Rab8A KO macrophages are significantly reduced after LLOMe treatment". How can Rab8A be phosphorylated in Rab8A KO cells?

Thanks for pointing this out. We have re-written this part to clarify what the experiments show.

The authors should check the citation format.

We have now formatting the citation format according to EMBO Journal requirements.

Referee #3:

The authors report for the first time that the LRRK2 kinase controls the repair of intracellular pathogen- (Mtb, Lm and Ca) or chemical-induced damaged endomembranes

(phagosomes and endolysosomes) through activating Rab8a and recruiting the ESCRT complex, thus preventing damaged endomembrane targeting to lysophagy. This is an important discovery in the field of cell biology and infection biology, that is supported by a nice combination of approaches, including cell imaging, and various pathogens. Importantly, the study also includes primary material (monocyte-derived macrophages) from patients carrying LRRK2 mutations.

The conclusions are well supported by the data. It would have been nice to study the behaviour of endomembranes in infected macrophages derived from PD patients' monocytes. Has this been done, please at least discuss.

This is a good point. Unfortunately, the samples were limited and only few macrophages could be differentiated from these donors. We are currently trying to recruit more patients in order to be able to do these experiments.

It is shown that Rab8-T72A is not recruited to the M. tuberculosis phagosome. Does this also hold true for Lm and Ca infected cells?

This is a very valid point. We have done these experiments and localization of Rab8A-T72A is shown for the three pathogens: Mtb, Lm and Ca (new Figure S3).

In addition to Rab8, Rab35 seems to show an increased recruitment to endolysosomes after damage; is that true, if so please discuss.

This is correct. We now discuss this in the text.

References to the figures in the first Results paragraph are inverted or mixed (Fig 2A instead of 1A, 2D-F instead of 1D-F, 2G-H instead of S2G-H etc.) Please fix.

Thanks for spotting this. We have now corrected it.

Thank you for submitting a revised version of your manuscript. It has now been seen by the original referees, whose comments are shown below.

As you will see, referee #3 finds that his/her criticisms have been sufficiently addressed and recommends the manuscript for publication. However, reviewer #1 and #2 still feel that some of their initial concerns remain valid. In particular, referee#1 states that experiments have to be properly quantified and analyzed statistically and that the issues about Rab specificity have not yet been solved. Referee #2 requests you to examine the kinetic of LRRK2 and Rab8 recruitment to damaged lysosomes at short time points using anti-LRKK2 antibody on fixed cells.

Given these comments from these two referees, I would like to invite you to perform the kinetic analysis of LRKK2/Rab8 recruitment to damaged lysosomes and proper statistical analysis suggested by referee #2 and #1, respectively. With regard to the issue about Rab specificity (reviewer #1), I urge you to openly acknowledge and discuss this limitation in the discussion section, and to avoid overstatements/overinterpretations especially in the title and abstract.

Please do not hesitate to contact me should you have any further questions. I look forward to your revision.

REFEREE REPORTS

Referee #1:

This revised manuscript provides evidence for a link between lysosomal injury and LRRK2 dependent phosphorylation of Rab proteins. The primary focus is on Rab8a. The major concern is that the submission fails to demonstrate a unique and special relationship between this one LRRK2 substrate as others and other rabs appear to be relevant as well. The reason for this concern is that lysosomal injury may simply activate large scale changes in intracellular trafficking such that most of the proteins one would study will appear affects, as is the case with most of the proposed negative controls that are not really negative (rab10, rab35) thus supporting this broad and non-specific view of the data. Thus, the mechanism-level detail one would expect for a high impact report in EMBO is not achieved. Rather, there are very interesting results that require additional follow up as the group concedes in the rebuttal, and controls this reviewer requested. Given the narrow window of time that these events were visualized for Rab8a one wonders whether most other rabs would show identical data, albeit with a unique temporal signature. Again, the use of the infection models in at least Rab8a, Rab10 and Rab35 KO cells would go a long way to show specificity in an open way where there is the opportunity to truly test their hypothesis, rather than collecting additional rab8a data to support it. Such efforts might begin to approach the level of rigor expected for this journal but the data [provided thus far indicate that is likely that each KO will show similar changes across the board.

Despite a borderline rejection ejection, the revised manuscript does not add sufficiently to the prior versions - stats are changed , but remain weak. The main new data in Fig 5 are troubling. First, they

are quantified over a single biological replicate as opposed to the three that were conducted, this is not appropriate. It would appear the group is counting individual cells for stats, instead of well-level or field-level averages, again, that have to be averaged across multiple biological replicates. While "statistically" different from a single experiment the differences between LRRK Wt and KO and GSK are artificially inflated by the use of cells (n=100s-1000s) as opposed to independent experiments (n=3) and one cannot see objectively a change.

Referee #2:

The authors have successfully addressed the points I raised, and I agree that some of my suggestions would be better suited for future studies. Overall I am very enthusiastic about this revised manuscript, but there is still one issue (raised in my original report) I think would be important to address, namely the kinetics of LRRK2 recruitment to damaged lysosomes. I fully understand that this is not feasible to do by live imaging, but the authors show several examples that their antibody against LRRK2 gives good staining by IF microscopy. The authors should thus examine the localization of LRRK2 (and Rab8A as a proxy) to lysosomes in cells fixed at short intervals after LLOMe treatment. I would suggest the following times: 0, 2, 5, 10 and 20 minutes.

Referee #3:

The authors have addressed my comments satisfactorily.

Point by point to reviewers**Referee #1:**

This revised manuscript provides evidence for a link between lysosomal injury and LRRK2 dependent phosphorylation of Rab proteins. The primary focus is on Rab8a. The major concern is that the submission fails to demonstrate a unique and special relationship between this one LRRK2 substrate as others and other rabs appear to be relevant as well. The reason for this concern is that lysosomal injury may simply activate large scale changes in intracellular trafficking such that most of the proteins one would study will appear affected, as is the case with most of the proposed negative controls that are not really negative (rab10, rab35) thus supporting this broad and non-specific view of the data. Thus, the mechanism-level detail one would expect for a high impact report in EMBO is not achieved.

We agree with the reviewer that other Rab GTPases could be participating in this process and we clearly stated that in the discussion. Here we show this is the case for - at least- Rab8A and our data with macrophages knockout for Rab8A (where other GTPases are expressed) clearly show that. There are likely different pathways activated after membrane damage, where several Rab GTPases could potentially be involved. We hope that our work will stimulate research primarily in this area, considering not much is known about how phosphorylation affects Rab protein function. We would like to highlight that we report here that the phosphorylation of Rab8A by LRRK2 is linked to endomembrane damage. Studying the phosphorylation of other Rab GTPases in macrophages will require to generate specific tools; e.g. antibodies that recognise the Rab and pRab in both Western blot and immunofluorescence. Another caveat is that not all Rab GTPases are expressed in macrophages and some are inducible. Many of these tools need to be generated and are beyond the scope of this study.

Rather, there are very interesting results that require additional follow up as the group concedes in the rebuttal, and controls this reviewer requested. Given the narrow window of time that these events were visualized for Rab8a one wonders whether most other rabs would show identical data, albeit with a unique temporal signature. Again, the use of the infection models in at least Rab8a, Rab10 and Rab35 KO cells would go a long way to show specificity in an open way where there is the opportunity to truly test their hypothesis, rather than collecting additional rab8a data to support it. Such efforts might begin to approach the level of rigor expected for this journal but the data [provided thus far indicate that is likely that each KO will show similar changes across the board.

We actually do not know if the whole subset of Rab GTPases that are eventually phosphorylated by LRRK2 will behave identically. In fact, our data with Rab35 shows that the phenotype is different from Rab8A (Figure EV5). Experiments with different knockouts are actually the beginning of these studies. Firstly, we need to identify what are the consequences of damage. Secondly understand if it is the Rab GTPase that it is involved in this process or the phosphorylation. Finally, these studies need to be done with the different pathogens and LLOMe and we do not know if they will all render identical results. Based on the evidence we provide, it will be unlikely that all the Rab GTPases phosphorylated by LRRK2 (around 14 Rab proteins when overexpressed,

Steger et al.,) behave identically and these studies are unfortunately beyond the scope of this study.

Despite a borderline rejection ejection, the revised manuscript does not add sufficiently to the prior versions - stats are changed, but remain weak. The main new data in Fig 5 are troubling. First, they are quantified over a single biological replicate as opposed to the three that were conducted, this is not appropriate. It would appear the group is counting individual cells for stats, instead of well-level or field-level averages, again, that have to be averaged across multiple biological replicates. While "statistically" different from a single experiment the differences between LRRK Wt and KO and GSK are artificially inflated by the use of cells (n=100s-1000s) as opposed to independent experiments (n=3) and one cannot see objectively a change.

We disagree with the reviewer, as per this reviewer suggestion we added a new complete figure showing the physiological relevance of this in the context of infection plus additional controls for the Western blots.

We also disagree with the remarks about inflation of the data. The use of high content microscopy allows us to see the data variation and distribution in a much robust way and count many more events, adding robustness to the analysis. In general, imaging analysis in macrophages show there is a large variability and counting limited number of events could lead to weak data. If anything, increasing the number of events make the analysis more powerful.

As suggested, we have now added into the graphs the mean of the 3 independent experiments (where at least 2.000 cells were counted) in **new Figure 4 and 5**. The conclusions remain the same as before.

Referee #2:

The authors have successfully addressed the points I raised, and I agree that some of my suggestions would be better suited for future studies. Overall, I am very enthusiastic about this revised manuscript, but there is still one issue (raised in my original report) I think would be important to address, namely the kinetics of LRRK2 recruitment to damaged lysosomes. I fully understand that this is not feasible to do by live imaging, but the authors show several examples that their antibody against LRRK2 gives good staining by IF microscopy. The authors should thus examine the localization of LRRK2 (and Rab8A as a proxy) to lysosomes in cells fixed at short intervals after LLOMe treatment. I would suggest the following times: 0, 2, 5, 10 and 20 minutes.

Many thanks for your positive comments. We have now performed the experiments. We analysed the recruitment of LRRK2, Rab8A, CHMP-4, Gal-3 and LC3B to damaged endolysosomes at the requested times (**new Figure 3, panel C and EV4**). This analysis shows that the localisation of LRRK2, Rab8A, CHMP4B, Galectin-3 and LC3B after endolysosomal damage is different. LRRK2 and Rab8A positive vesicles were visible as early as 10 min after LLOMe stimulation. The occurrence of LRRK2 and Rab8A positive vesicles coincided with CHMP4B but not Galectin-3 or LC3B positive vesicles which were visible after 15 min after LLOMe treatment (**Fig. 3C and Fig. EV4B**).

Thank you for submitting a revised version of your manuscript. It has now been seen by referee #2, whose comments are shown below.

As you will see, this reviewer finds that his/her remaining criticisms have been sufficiently addressed and recommends the manuscript for publication. With regard to the concerns on which Rabs are involved in LRKK2-mediated membrane repair and the mechanistic insight thereof raised by referee #1, we find that these issues do not preclude publication here.

Before we can officially accept your manuscript, there are a few editorial issues concerning text and figures that I need you to address.

REFEREE REPORTS

Referee #2:

The authors have successfully addressed the remaining point I raised, and I am happy to recommend publication of this revised manuscript in EMBO Journal.

Corresponding Author Name: Maximiliano Gutierrez

Manuscript Number: EMBOJ-2020-104494R2